# Large to submesoscale surface circulation and its implications on biogeochemical/biological horizontal distributions during the OUTPACE cruise (SouthWest Pacific)

Louise Rousselet[1], Alain de Verneil[1], Andrea M. Doglioli[1], Anne A. Petrenko[1], Solange Duhamel[2], Christophe Maes[3], and Bruno Blanke[3]

[1]Aix Marseille Univ, Universite de Toulon, CNRS, IRD, OSU PYTHEAS, Mediterranean Institute of Oceanography MIO, UM 110, 13288, Marseille, Cedex 09, France
[2]Lamont-Doherty Earth Observatory, Division of Biology and Paleo Environment, PO Box 1000, 61 Route 9W, Palisades, NY 10964, USA
[3]Laboratoire d'Oceanographie Physique et Spatiale, CNRS, Ifremer, IRD, UBO, Brest, France

*Correspondence to:* Rousselet Louise (louise.rousselet@mio.osupytheas.fr)

**Abstract.** The patterns of the large-scale, meso- and submesoscale surface circulation on biogeochemical and biological distributions are examined in the Western Tropical South Pacific (WTSP) in the context of the OUTPACE cruise (Feb-April 2015). Multi-disciplinary original *in situ* observations were achieved along a zonal transect through the WTSP and their analysis was coupled with satellite data. The use of Lagrangian diagnostics allows for the identification of water mass pathways, mesoscale structures, and submesoscale features such as fronts. In particular, we confirmed the existence of a global wind-driven southward circulation of surface waters in the entire WTSP, using a new high-resolution altimetry-derived product, validated by *in situ* drifters, that includes cyclogeostrophy and Ekman components with geostrophy. The mesoscale activity is shown to be responsible for counter-intuitive water mass trajectories in two subregions: i) the Coral Sea with surface exchanges between the North Vanuatu Jet and the North Caledonian Jet; and ii) around 170°W with an eastward pathway whereas a westward general direction dominates. Fronts and small-scale features, detected with Finite-Size Lyapunov Exponents (FSLE), are correlated with 25% of surface tracer gradients which reveals the significance of such structures in the generation of submesoscale surface gradients. Additionally, two high-frequency sampling transects of biogeochemical parameters and microorganism abundances demonstrate the influence of fronts in controlling the spatial distribution of bacteria and phytoplankton, and as a consequence the microbial community structure. All circulation scales play an important role that has to be taken into account when analysing the data from OUTPACE but also, more generally, to understand the global distribution of biogeochemical components.

## 1 Introduction

The zonal trophic gradient of the Western Tropical South Pacific (WTSP) represents a remarkable opportunity to study the interactions between marine biogeochemical Carbon (C), Nitrogen (N), Phosphorus (P), Silica (Si), and Iron (Fe) cycles between different trophic regimes. One of the OUTPACE cruise's goals is to understand how $N_2$ fixation controls production, miner-

alisation and export of organic matter (Moutin and Bonnet, 2015; Moutin et al., 2017). The ocean's circulation, at different time/space scales, can play a key role in biological variability and dynamics. In particular the meso- and submesoscale, which occurs at scales typical of phytoplankton blooms (Dickey, 2003), enhance carbon export through vertical motion (Guidi et al., 2012; Lévy et al., 2012) and thus strongly impact the biological pump.

Indeed, mesoscale dynamics (features with time/space scales on the order of months/100 km such as eddies) can affect biological and biogeochemical cycling through transport processes such as horizontal advection, lateral stirring and eddy trapping, as well as through processes that modify nutrients and/or light availability such as eddy pumping, eddy-wind interaction or frontal instabilities (Williams and Follows, 1998; McGillicuddy Jr, 2016). Some observations, mostly collected during the JGOFS program, have shown the influence of eddy circulation in sustaining primary production in oligotrophic regions (Jenk-

ins, 1988; McGillicuddy and Robinson, 1997). Numerically modelled eddy fields also show an enhancement of biological productivity in most provinces of the North Atlantic Ocean (Oschlies and Garcon, 1998; Garçon et al., 2001). Additional studies also discuss the structuring effect of mesoscale features, such as vortices, on ecological niche composition and distribution, depending on eddy characterictics and eddy stirring (Sweeney et al., 2003; d'Ovidio et al., 2010; Perruche et al., 2011; d'Ovidio et al., 2013). Therefore mesoscale features can have strong ecological impacts through the enhancement of biological

production and the creation of favourable conditions for less competitive species, with implications for higher trophic levels.

    Smaller scale dynamics may also have a significant role in the distribution of biological variability. The submesoscale represents the ocean processes characterized by horizontal scales 1-10 km which origins might be linked with the stirring induced by mesoscale interactions and frontogenesis (Capet et al., 2008). At this typical scale, the flow can be characterized by strong stretching lines or vortex boundaries creating physical barriers such as fronts or filaments that are associated with sharp gra-

dients. These structures can contribute to the separation or mixing of water masses and thus impact the horizontal distribution of tracers, biogeochemical and biological matter such as biomass of phytoplankton cells at a front boundary. Indeed microorganisms are buoyant material and their distribution, as well as the biogeochemical components, can be driven by submesoscale activity, whether through direct horizontal advection (Dandonneau et al., 2003), or indirectly following the biogeochemical dynamics. Nitrogen fixing organisms such as *Trichodesmium spp.*, which contribute in sustaining high primary productivity

in the Pacific ocean, are known to concentrate around small-scale features in the North Pacific (Fong et al., 2008; Church et al., 2009; Guidi et al., 2012). In the WTSP, Bonnet et al. (2015) argued that *Trichodesmium spp.* abundances might follow gradient distributions. Besides regulating the spatial distribution of microorganisms, these features can participate in biological dynamics as they can induce vertical movments of nutrient supplies and chlorophyll (Martin et al., 2001). Lévy et al. (2015) showed that the flow field brings populations into contact in frontal areas which can be characterized by a larger diversity

of microorganisms and more fast-growing species. Consequently, submesoscale circulation can influence the planktonic community structure. However, due to their typical scales, mesoscale and submesoscale features, require substantial means to be adequately observed (Mahadevan and Tandon, 2006) and their interactions with biogeochemistry and biology are also hard to elucidate due to their ephemeral nature (McGillicuddy Jr, 2016).

    In the WTSP the large scale circulation is dominated by the anticylonic South Pacific Gyre. The South Equatorial Current

(SEC) flows in the equatorial band (0°S - 6°S) from East to West and is divided in multiple branches when approaching the

Coral Sea (Webb, 2000; Sokolov and Rintoul, 2000) (Figure 1). On the western boundary, the East Australian Current (EAC) feeds, through the Tasman Sea, the southern branch of the gyre which then flows east and reaches the Peru/Chile Current (PCC) near the western South American coast (Tomczak and Godfrey, 2013). Superimposed on these large scale patterns, several studies indicate a strong mesoscale variability due to barotropic and baroclinic instabilities and to the interactions of the major currents and jets with the numerous islands of the region (Qiu and Chen, 2004; Qiu et al., 2009; Hristova et al., 2014). As displayed in Figure 1, the OUTPACE cruise was conducted in the transition area between a zonal band of relatively high eddy kinetic energy south of 19°S (Qiu et al., 2009) and low eddy kinetic energy to the north. The influence of this intense variability, which results in mostly westward propagating eddies (Chelton et al., 2007; Rogé et al., 2015), has not been fully explored yet (Kessler and Cravatte, 2013), as well as its implications on biogeochemical/biological variations in the region. Recent studies have underlined the role of mesoscale activity as a conveyor of water masses, leading to the discovery of a potential water mass pathway in the Coral Sea (Maes et al., 2007; Ganachaud et al., 2008; Rousselet et al., 2016). This intense mesoscale activity is strongly linked to submesoscale fronts that might be responsible for surface small-scale features in temperature and salinity as shown by Maes et al. (2013) within the Coral Sea. Submesoscale dynamics are also thought to be responsible for ∼20% of new production in oligotrophic regions as suggested by Lévy et al. (2014b) using an idealized model. Since the frequency of oceanic fronts and eddy kinetic energy should increase in oligotrophic regions with climate change (Matear et al., 2013; Hogg et al., 2015) the OUTPACE cruise offers an unprecedented opportunity to study large, meso- and submesoscale influences along a zonal gradient crossing the oligotrophic to ultra-oligotrophic WTSP with coupled physical and biogeochemical measurements.

In this study we place the *in situ* observations collected during the OUTPACE cruise into a synoptic view of the WTSP circulation at different horizontal scales. We investigate, through a descending approach, the large, meso- and submesoscale patterns using *in situ* observations, coupled with satellite data. Remote sensing provides daily physical and biological information over the entire WTSP for a time period covering the cruise duration and beyond (from June,1 2014 to May,31 2015). The inter-comparison between physical lagrangian diagnostics and available biogeochemical/biological measurements explores the potential influence of each scale on biogeochemical variations. In particular, we propose to focus on the possible impacts of horizontal small-scale ocean circulations on horizontal distribution of temperature, salinity or chlorophyll, as well as on surface phytoplankton. Two original case studies are also presented to illustrate the fine-scale physical influence on horizontal microbial distributions. The use of multidisciplinary approaches, including *in situ* observations, remote sensing and numerical simulations is the key aspect of this study to investigate the surface circulation at different scales and try to examine their potential influence on the distribution of biogeochemical parameters and major groups of plankton measured during the OUTPACE cruise. In the following, we present the datasets and methods used and we discuss the results for each circulation scale, from large to submesoscale.

## 2 Materials and Methods

### 2.1 *In situ* observations

The OUTPACE (Oligotrophy to UlTra-oligotrophy PACific Experiment) cruise performed a zonal transect across the WTSP aboard R/V *L'Atalante* from February 18, 2015 to April 3, 2015 (Moutin and Bonnet, 2015). The main objectives of the cruise were to study the interactions between planktonic organisms and the cycling of biogenic elements across trophic and $N_2$ fixation gradients (Moutin et al., 2017). A total of 15 hydrological stations were sampled along the transect as well as three long-duration (LD) stations named LDA, LDB and LDC (Fig. 1). LD station sampling lasted for almost 8 days each, and aimed to study carbon export in 3 biogeochemically different regions. More details about the sampling strategy are available in Moutin et al. (2017). The multi-disciplinary measurements used in this study are described hereinafter.

Of particular interest to understand the surface dynamics, SVP (*Surface Velocity Program*) floats were launched during each LD station to investigate the dispersion and the surface circulation relative to 15 m during the sampling period and beyond (Lumpkin and Pazos, 2007). Three SVPs were launched during LDA, 6 during LDB and 4 during LDC. The *in situ* trajectories of the floats were used to validate altimetry-derived surface velocities (see Sec. 2.2, 2.4 and Fig.2).

Continuous measurements of temperature and salinity were achieved using a ThermoSalinoGraph (TSG) that pumped sea water at 5 m depth. TSG data have been corrected and calibrated using independent measurements of salinity from water bottle samples collected daily onboard R/V *L'Atalante* (following the procedures described by Alory et al. (2015)) and are binned into minutes. In the following, temperature and salinity will refer to absolute salinity and conservative temperature, respectively, according to TEOS-10 standards (McDougall et al., 2012). A Wetstar SeaBird fluorimeter was deployed on the underway water flow. The fluorimeter provides measurements proportional to the chlorophyll a concentration with a time step of 10-15 min. Discrete samples were taken during the transit to calibrate the fluorimeter using the Aminot and Kérouel (2004) method:

$$\text{Chla [mg m}^{-3}] = 1.99 \times \text{FluorescenceValue} - 0.083 \ (R^2 = 0.87, n = 55)$$

Due to technical issues the underway sampling of chlorophyll concentration started on March 7. Each of these data sets have been interpolated on a regular grid of 0.5 km resolution in order to keep the high resolution, but equally distributed along the travelled distance.

Two high frequency samplings (every 20 min) were performed: the first one upon leaving LDA and the second one upon arriving at LDB, in order to assess variability in 15 different biogeochemical parameters, in particular the Dissolved Inorganic phosphate (DIP) turnover times and the abundances of bacteria (including the low and high nucleic acid content bacterial groups, LNA and HNA, respectively), *Prochlorococcus*, *Synechococcus* and picophytoeukaryotes (PPE). Dissolved inorganic phosphate turnover times (TDIP) were determined using a dual $^{14}$C - $^{33}$P labelling method following Duhamel et al. (2006). As described in Moutin et al. (2017), TDIP represents the ratio between phosphate natural concentration and phosphate uptake

by planktonic species (Thingstad et al., 1993). It is considered the most reliable measurement of phosphate availability in the upper ocean waters (Moutin et al., 2007). In the WTSP, the phytoplankton growth is often limited by phosphate availability. Consequently, this parameter gives important information on the biological activity in relation to resource availibility : a very short TDIP means rapid utilization of the ambient phosphate present in limiting concentration. To enumerate cell abundances of these different microbial groups, water samples were collected directly from the underway pump, fixed with 0.25% (w/v) paraformaldehyde, flash frozen and preserved at -80°C until analysis by flow cytometry following the protocol described in Bock et al. (2018). Briefly, bacteria were discriminated in a sample aliquot stained with SYBR Green I DNA dye (1:10,000 final) while pigmented groups were discriminated in an unstained sample aliquot. Reference beads (Fluoresbrite, YG, 1 $\mu$m) were added to each sample. Particles were excited at 488 nm (plus 457 nm for unstained samples) and bacteria were discriminated based on their green fluorescence and forward scatter (FSC) characteristics, while *Prochlorococcus*, *Synechococcus* and PPE were discriminated based on their chlorophyll (red) fluorescence and FSC characteristics. LNA and HNA groups were further distinguished based on their relatively low and high SYBR Green fluorescence, respectively, in a green fluorescence vs side scatter plot. *Prochlorococcus* were further distinguished from *Synechococcus* by their relative lack of a phycoerythrin signal (orange fluorescence). Using a FSC detector with small particle option and focusing a 488 plus a 457 nm (200 and 300 mW solid state, respectively) laser into the same pinhole greatly improved the resolution of dim surface *Prochlorococcus* population from background noise in unstained samples. Because the *Prochlorococcus* population cannot be uniquely distinguished in the SYBR stained surface samples, bacteria were determined as the difference between the total cell numbers of the SYBR stained sample and *Prochlorococcus* enumerated in unstained samples. Cytograms were analyzed using FCS Express 6 Flow Cytometry Software (De Novo Software, CA, US). These data are used to investigate the small-scale distribution of the different microbial community groups and its relation with the concomitant dynamics at submesoscale. To investigate each picoplankton group variability with respect to the other with more clarity, abundances are displayed in terms of cell count deviation. It represents the difference between picoplankton group abundance (cell mL$^{-1}$) measured at a certain location and the mean of the respective picoplankton group abundances throughout the transect.

## 2.2 Satellite data

Several satellite datasets were exploited during the campaign to guide the cruise through an adaptive sampling strategy using the SPASSO software package (http://www.mio.univ-amu.fr/SPASSO/) following the same approach as described for previous cruises such as LATEX (Doglioli, 2013; Petrenko et al., 2017) and KEOPS2 (d'Ovidio et al., 2015). SPASSO was also used after the cruise in order to extend the spatial and temporal vision of the *in situ* observations.

Four different altimetry-derived velocity products were tested in this study to choose the product that best represents the surface circulation during the cruise. First the daily Ssalto/Duacs product (Ducet et al., 2000), from AVISO (Archiving, Validation and Interpretation of Satellite Oceanographic 3) data base, for the period from 1 January 2004 to 31 December 2015, was used to extract daily delayed-time maps of absolute geostrophic velocities (1/4° x 1/4° on a Mercator grid, since 15 April 2014).

Three other altimetry products were specifically produced, for the first time, at 1/8° resolution for the WTSP region by Ssalto/Duacs and CLS (Collecte Localisation Satellites), with support from CNES (Centre National d'Études Spatiales), from June 2014 to June 2015. In particular, they provided daily maps of absolute geostrophic velocities, daily maps of the sum of absolute geostrophic velocities and Ekman components, and the same product as the latter that also includes a cyclogeostrophy

correction (Penven et al., 2014). Ekman surface currents refer to the wind-induced circulation relative to 15 m and are computed from ECMWF ERA INTERIM windstress with an Ekman model fitted onto drifting buoys (Rio et al., 2014).

A preliminary comparison between Lagrangian numerical particle trajectories (see Sec. 2.3.1), computed with each of the above products, and the observed trajectories of the different floats launched during OUTPACE (Sec. 2.1) allowed us to identify the product including geostrophic and Ekman components with the cyclogeostrophy correction (hereafter total altimetry-derived

velocity field) as the most accurate *in situ* surface currents (see Sec. 2.4).

Daily near-real-time maps of sea surface temperature (SST) and ocean color were also specifically produced for the WTSP from December 2014 to May 2015 by CLS with support from CNES. They are constructed with a simple weighted data average over the 5 previous days (giving more weight to the most recent data), and have a 1/50° resolution (2 km at the Equator) in latitude

and longitude. The temperature product corresponds to maps of SST deduced from a combination of several intercalibrated infrared sensors (AQUA/MODIS, TERRA/MODIS, METOP-A/AVHRR, METOP-B/AVHRR). The ocean color product corresponds to maps of chlorophyll concentration issued from the Suomi/NPP/VIIRS sensor (http://npp.gsfc.nasa.gov/viirs.html). These satellite data are compared with *in situ* data from the underway survey. A correlation of 0.8 between *in situ* measurements and co-located satellite data validates the satellite-derived SST and chlorophyll concentration. A supplementary

correlation with the daily High-Resolution SST blended from NCDC/NOAA (Reynolds et al., 2007) and *in situ* SST showed a similar correlation. These results corroborate the accuracy of the CLS products in our region of interest.

## 2.3  Lagrangian diagnostics

### 2.3.1  Surface water mass pathways detection

To investigate the water mass movements at large and meso-scale, we used the Lagrangian diagnostic tool Ariane that can

trace water mass movements from the trajectories of numerical particles that enter and exit a predefined domain (Blanke and Raynaud, 1997; Blanke et al., 1999). In this study the numerical particle trajectories are computed with altimetry-derived surface currents from the products listed above (see Sec. 2.2). As many Lagrangian particles as desired can be integrated in two different ways: backward in time to assess the origins of the water masses or forward in time to investigate their fate. Additionally, this Lagrangian tool allows for the computation of two different diagnostics : i) qualitative diagnostics that compute

typically few particles with a steady recording of the positions along their trajectories; ii) quantitative diagnostics that compute thousands of particles with statistics available for initial and final positions, and with the diagnostic of the main pathways. In this study we use 3 different configurations of the Ariane tool depending on the objectives.

First, to identify the altimeter product that best fits the observed trajectories of the floats launched during OUTPACE, a com-

parison is performed, in the following section 2.4, with the trajectories of one hundred numerical particles. They were initially positioned around the launching position of the floats with a resolution of 1-2 km.

The particles were advected forward in time for 96 (LDA), 78 (LDB) and 70 (LDC) days, corresponding to the time lapse between the launch day of the floats and the last available day of satellite data. These qualitative experiments allows for the comparison of successive positions (every 6 hours) of numerical particles computed with the 4 different products and those of the floats. Thus the choice of the best surface velocity product relied on the best fit between observed and numerical trajectories (see Sec. 2.4).

To study the large scale circulation in the WTSP, quantitative experiments were performed to find the main pathways of the waters entering and exiting the box contouring the WTSP (Fig. 3). Particles were launched along each section of the box (North, East, South, West) and advected forward with the total altimetry-derived surface currents. We simulated ten years of particle trajectories by repeating ten times the available dataset. We compared the results of this simulation with the ten years integration of available geostrophic AVISO surface currents over this time period. The comparison between these simulations ensures that the use of a one year time period looped several times does not significantly modify statistical outputs. It is clearly more interesting to use CLS data instead of commonly used AVISO geostrophic surface currents because they include the wind effect and cyclogeostrophy with higher resolution, as well as better fitting with *in situ* data.

Another objective is to identify the mesoscale trajectories of surface water masses sampled at each of the LD stations. Backward and forward quantitative experiments were performed to identify the main pathways of the surface water masses arriving and leaving each LD station using total altimetry-derived surface currents. Almost one million numerical particles were initially distributed along a square box surrounding the position of the LD station. The calculations were stopped whenever the particles return to the initial box (hereafter called meanders) or were intercepted on one of the four remote sections located around the LD station. The boxes size are tuned in order to minimize meanders and the loss of particles in the domain. Percentage of both quantities are reported in Table A1. Forward computation times are identical to the qualitative diagnostics. For backward experiments, the particles were advected for 183 (LDA), 201 (LDB) and 209 (LDC) days corresponding to the maximum time lapse allowed by CLS satellite data availability.

### 2.3.2   Eddies and filaments identification

To set up the mesoscale context during the OUTPACE cruise, we used the Lagrangian Averaged Vorticity Deviation method (Hadjighasem and Haller, 2016; Haller et al., 2016) that allows identification of coherent structures from altimetry-derived surface velocity fields (code available at https://github.com/Hadjighasem/Lagrangian-Averaged-Vorticity-Deviation-LAVD). The detected features are able to trap water masses for a certain period (defined by the integration time) and transport them along their route. In this study we computed the detection with the total altimetry-derived velocity field and chose an 8 day time integration with respect to the duration of LD stations. Indeed this time interval provides a confirmation or rebuttal of the assumption that LD stations have been performed in a coherent structure, as targeted during the cruise. This Lagrangian diagnostic is also compared with a hybrid Lagrangian and Eulerian approach combining the calculation of the Okubo-Weiss (OW) parameter and a retention parameter (RP), computed with the same velocity field. The OW parameter identifies structures

such as eddies by separating the flow into a vorticity-dominated region and a strain-dominated region (Okubo, 1970; Weiss, 1981). The RP identifies the number of days a fluid parcel remains trapped within a structure core, defined by a negative OW parameter. Both parameter calculations are detailed in d'Ovidio et al. (2013). As for the LAVD detection method, the RP allows for the identification of potentially trapping coherent structures.

Submesoscale flow features in two-dimensional data are evaluated with altimetry-derived finite size Lyapunov exponents (FSLE), computed with the algorithm of d'Ovidio et al. (2004). This Lagrangian diagnostic detects frontal zones on which passive elements of the flow should theoretically align. Here we used the total altimetry-derived velocity field to compute the algorithm. The main parameter values for the algorithm are described in de Verneil et al. (2017b). The OUTPACE cruise occurred in the relatively open ocean and far enough from islands to trust the FSLE diagnostic calculated from altimetry. We compare the horizontal positions of the fronts, detected with FSLE, with surface gradients measured both with the TSG (temperature and salinity) and the underway survey (chlorophyll). Indeed these high frequency samplings provide access to submesoscale gradients. A point by point correlation is calculated whenever a strong gradient of density (or chlorophyll) corresponds or not to a high FSLE value (i.e. $> 0.05$ day$^{-1}$) indicative of a front. Sensitivity tests have been performed to choose the thresholds on density (chlorophyll) gradients to ensure the stability of the correlations calculated. These tests ended with the selection of gradient larger than 0.1 kg m$^{-3}$ km$^{-1}$ and a 0.2 mg m$^{-3}$ km$^{-1}$ as thresholds for density and chlorophyll gradients, respectively.

## 2.4 Comparison of satellite products with *in situ* drifters

The choice of the satellite product that best represents the surface dynamics relies on a qualitative comparison between the trajectories of *in situ* floats launched during the OUTPACE cruise (see Sec. 2.1) and the trajectories of numerical particles computed with each of the satellite-derived velocity field described in Section 2.2. Figure 2 shows the 8-days trajectories of in situ floats and numerical particles at each long-duration station and for each satellite-derived products considered (geostrophy 1/4°; geostrophy 1/8°; geostrophy and Ekman 1/8°; geostrophy, Ekman and cyclogeostrophy 1/8°). The comparison is restricted to 8 days for a better visualisation and to be consistent with the duration of the LD stations. In the case of station LDA, none of the products displays a significant improvement of numerical trajectories. This lack of refinement between the different products may be due to the lack of accuracy of satellite products when getting close to coasts. Indeed, altimetry measurements are not well resolved close to the coast, and especially near New Caledonia where the topography and bathymetry are very complex. In the cases of LDB and LDC, the increase in resolution does not modify the general pattern of the trajectories. However, when adding the Ekman component, we can notice an improvement in the direction of the numerical particle trajectories. Even if the particle positions are offset, their direction are consistent with those of in situ drifters. Cyclogeostrophy seems to accelerate the particles' displacements. The final positions of the numerical particles are closest to the final position of in situ drifters in the case of LDC. This latter point is not surprising considering that cyclogeostrophy represents the centrifugal acceleration. In the context of the OUTPACE cruise, we consider that the LDB and LDC examples illustrate clear improvements of the new satellite product including geostrophy, the Ekman component and cyclogeostrophy. In the case

of LDA neither clear improvement or deterioration are obvious on the trajectories. Moreover when considering the surface circulation, it also remains important to take into account the wind effect, through the Ekman component, as it will strongly influence the trajectories of surface waters at large, meso- and submesoscale. As most of the diagnostics used in this study are calculated through particle trajectory computations, the Ekman component is of major significance. As a consequence, and to stay consistent throughout this study, we used the product combining geostrophy, the Ekman component and cyclogeostrophy at 1/8° resolution, referred as the total surface altimetry-derived velocity field, to compute every diagnostic.

## 3 Results and discussions

### 3.1 Large scale wind-driven pathways

The geostrophic large scale mean circulation and directions of the main currents in the WTSP are well established from the literature. However, the trajectories and pathways of surface waters may change and be more complex when the effect of the wind is added and resolution is increased especially in the context of inter-annual ENSO (El Niño Southern Oscillation) variability. We decided to use a Lagrangian integration of numerical particles to simulate the transport of surface fluid parcels at the scale of the WTSP region using altimetry-derived ocean currents including the wind effect. Figure 3 shows the transport calculated from the sum of almost 13 million numerical particles advected with the total surface altimetry-derived flow for 10 years (see Sec. 2.3.1). This figure highlights the westward transport of the SEC in the northwestern part of the domain, but also the eastward transport at 10°S due to the output of the South Equatorial Counter Current (SECC) from the Solomon Sea (Ganachaud et al., 2014). Both these pathways follow the well-known circulation of the SEC and SECC in this region. An eastward flow south of Fiji is also detectable on Figure 1 but does not seem to influence surface transport (Fig. 3). This flow has been discussed in several studies and is named the South Tropical Countercurrent (STCC) (Qiu and Chen, 2004). Additionally, a clear surface meridional transport is noticeable from 10°S to 25°S. In the very southeastern part of the domain some surface waters seem to recirculate to the east from 170°W, which is probably an indicator of the gyre circulation. The meridional transport does not correspond to any general surface geostrophic current previously described in the literature (Tomczak and Godfrey, 2013; Kessler and Cravatte, 2013; Ganachaud et al., 2014), but is mainly due to the south-easterly trade winds. This meridional component appears due to the addition of the Ekman component in altimetry surface velocities (Fig.A1, Supplementary Material). The large scale transport of surface waters in the OUTPACE area is thus a combination of the transport by general well-known surface currents and wind-driven circulations. Most of the surface waters travel southwest from the northern Ariane section, with a significant part that originates from northeast. At the scale of the WTSP, the individual transport calculated from each initial section (see Sec. 2.3.1) reveals that 80% of the surface waters crossed during the OUTPACE cruise originate from the «North» section, 8-15% from the «East» section and very few from the «South» and «West» sections (Fig. A2). We can thus identify a general wind-driven surface transport in the WTSP as follows: the surface waters enter the WTSP from the northeast with the SEC and are gradually advected to the south with a part (east of 170°W) that directly recirculates within the gyre and another part (west of 170°W) that follows a southwestern propagation through the different WTSP archipelagos. These results

obtained with a Lagrangian diagnostic complete the largely elucidated eulerian vision of the large scale circulation in the WTSP.

Very few reports have studied the surface transport inferred by the wind in the WTSP. Indeed Tomczak and Godfrey (2013) calculated a streamfunction from wind stress and showed a global westward transport with very little meridional transport ex-
cept in the EAC which is very close to the Australian coasts. Kessler and Cravatte (2013) also computed the Sverdrup transport streamfunction calculated from Godfrey's Island Rule and the wind stress curl field in the Coral Sea. They identified a comparable southward transport as visible in Figure 3, between 7°S and 12°S, due to the SECC. However, between 12°S and 25°S, they identified a westward transport whereas we show a meridional transport from north to south. Considering the large scale biogeochemical distribution, two types of waters can be differentiated: the relatively oligotrophic but richer Melanesian waters
(from 160°E to 170°W) and the ultra-oligotrophic gyre waters (east of 170°W) (Fumenia et al., 2018). The wind-driven surface transport highlighted here could participate in the biogeochemical variations between western and eastern waters. Moreover, the path through the Melanesian area, which includes the multiple islands from Papua New Guinea to Fiji (140°E-170°W), may enrich these waters due to the contact with multiple islands. This could explain the relatively higher productivity of these waters, whereas waters that directly recirculate within the gyre keep their ultra-oligotrophic characteristics.

The WTSP circulation is also strongly impacted by ENSO conditions, responsible for SST variability on inter-annual to decadal timescales (Sarmiento and Gruber, 2006). A negative Southern Oscillation Index (SOI), El Niño phase, is characterized by a decrease or even an overturn of trade winds whereas during La Niña phase (positive SOI) trade winds are strengthened. The mean wind velocity measured during OUTPACE is shown to be close to mean velocities during El Niño (data not shown)
and Moutin et al. (2017) clearly showed that the OUTPACE cruise took place during an El Niño phase but they determined that climatological effects, upon the results of the cruise, were minimized for biogeochemical sampling. However we show that the circulation and transport are still strongly influenced by the trade winds.

As this region is characterized by an intense mesoscale circulation, we can also expect that it participates in the biogeochemical variations in the region. Thus we investigate the mesoscale feature trajectories on the entire WTSP and in particular
the mesoscale circulation around three biogeochemically different locations: i) LDA located in the Coral Sea, at the end of the surface waters' journey across the WTSP; ii) LDB, in the Melanesian waters, west of the transition zone with gyre waters; iii) LDC, in the gyre waters.

## 3.2 Mesoscale activity and trajectories of surface waters

The major goal in this section is to identify whether mesoscale activity, and in particular trapping features, actively participate
in the transport of different water mass properties across the WTSP ocean. The LAVD method is chosen in order to track the coherent structures for a time period of 8 days (see Sec. 2.3.2). Figure 4 shows the total altimetry-derived velocity field and the mesoscale structures identified for the first day of each LD station (LDA, LDB and LDC). It reveals that many mesoscale structures are detected by the LAVD method in the entire WTSP with no specific region with higher abundances of these features. To ensure that this recent detection method is consistent with previous approaches that identify mesoscale features (eulerian

Okubo-Weiss parameter) or retention areas (RP), we compare the structures detected with both approaches. A comparison with the OW parameter method shows a good agreement with the LAVD detection method. Indeed, a mean OW parameter value of $-0.24$ day$^{-2}$ is calculated inside the contour of coherent structures detected with the LAVD method. It indicates that wherever a coherent structure is identified, the OW parameter also identifies a mesoscale feature. Most of the mesoscale structures detected

with the LAVD method are also identified as retention areas, with the RP, ensuring the trapping characteristics of these features (Fig. A3). A tracking of these coherent structures highlights that they all show a general westward propagation as expected from the mean circulation in this area.

Coherent mesoscale features are well-known to participate in the surface biogeochemical variations through eddy trapping and transport. Unfortunately, in our case, we are not able to observe the trapping and transport of different water masses

by the mesoscale structures with *in situ* data. Indeed the zonal equidistant biogeochemical sampling during OUTPACE did not sample both mesoscale features and surrounding waters. Consequently no differentiation is possible between potentially trapped waters and surrounding waters. The small differences between water mass properties in this region (Gasparin et al., 2014) may also make it difficult to confidently notice a biogeochemical marker of different water masses. Even if, in this case, the influence of mesoscale activity, through eddy trapping and transport, on biogeochemical variations is not directly visible on

*in situ* data, the role of mesoscale dynamics on the trajectories of surface waters can document the possible exchanges between biogeochemically differentiated regions.

Here we also dynamically explore the origins and fates of surface waters sampled during each LD station located in three different environments. LDA is situated on the path of the westward North Caledonian Jet (NCJ), that flows between New Caledonia and Vanuatu, in relatively highly productive waters (Fig.6). Far to the east, LDB is positioned near the limit between

oligotrophic and ultra-oligotrophic waters inside a phytoplankton bloom whereas LDC is located in nutrient-poor waters in the South Pacific (SP) gyre (Fig.6). Figure 5 shows the streamfunctions calculated from numerical particle advection using total altimetry-derived surface velocities (see Sec. 2.3.1). They represent the origin (Fig. 5 top) and the fate (Fig.5 bottom) of each LD stations' waters, respectively the backward and forward computations. One would expect LDA surface waters to come from the East as it is located on the path of the westward NCJ. However they seem to have multiple origins: i) easterly, directly

from the NCJ transport; ii) northerly, directly from waters that have circulated between the Vanuatu islands before heading south to LDA; and iii) from a meridional tortuous recirculation path ($\sim$162°E) within the Coral Sea. After LDA sampling, an intense signature of the NCJ is detected at the surface. Another portion of surface waters directly crash on New Caledonia's northern coast. In the eastern WTSP, LDB surface waters seem to follow the same general path: they flow from northeast towards southwest before they reach a group of islands and then recirculate to the east towards LDB. After LDB sampling,

they continue their way to the east before heading back to the south or to the south-west. Further east, as one would expect, the waters sampled during LDC travelled from the east and flow to the west after LDC (Fig.5). We notice a recirculation area east of LDC where the waters seemed to follow a looping trajectory before reaching LDC. Both lagrangian methods (LAVD detection and advection of numerical particles) detect a coherent mesoscale structure that travelled westward in the surrounding region of LDC.


The identification of coherent structures during OUTPACE revealed that only station LDC could be influenced by a trapping structure. A tracking of this structure, as well as the high rates of meanders (70% and 44% for backward and forward integration, respectively) (Table A1), suggest that this coherent structure crossed the LDC sampling area. Indeed some CTD casts were performed inside or near the boundary of this structure whereas others, mostly at the end of the station were realised

after the crossing of the structure. This observation can be associated with the results of de Verneil et al. (2017a) who identified, with *in situ* data, a modification in the water mass composition throughout the station. We thus suggest that the change in the physical environment during LDC could be due to the westaward transit of this coherent structure across LDC sampling site. If only LDC sampling site seems to be directly influenced by the transport of water masses through a coherent mesoscale structure, the trajectories at LDA and LDB highlight some interesting mesoscale path (circulation at the scale of the order of

10-100 km). Around LDA, both i) and ii) origins agree with the integrated transport entering the Coral Sea induced by the complex topography (Kessler and Cravatte, 2013). The westward circulation scheme is consistent with previous studies focusing on the NCJ (Ganachaud et al., 2008; Gasparin et al., 2011) and more consistently with the results analyzed by Barbot et al. (2018) within the same context and cruise. The pathway that crashes on New Caledonia's coast might not be so relevant due to the satellite's lack of resolution near coastal areas. The meridional recirculation previously determined suggests that eddy-eddy

interactions might be responsible for the emergence of complex paths between the NVJ and the NCJ. Moreover backward and forward streamfunctions cross around station LDA which suggests that the area between New Caledonia and Vanuatu is a region of complex recirculation with waters that stay in this region for a while before exiting the Coral Sea. These observations match the area described by Rousselet et al. (2016) as the region of exchange between NCJ and NVJ waters through eddy trapping and transport. We identify a probable water mass mixing area, in the Coral Sea center, through complex mesoscale

stirring. This stirring may also create surface gradients as depicted by Maes et al. (2013). Around LDB, an eastward path is detected that could match with the eastward flow of the STCC. Moreover de Verneil et al. (2017b) also pointed out a possible eastward transport to explain the origin of the bloom sampled at LDB. Indeed the surface waters might be iron-enriched through contact with the islands and thus create favourable conditions for a phytoplankton bloom. At this site, adjacent to the nutrient-poor SP gyre, the biological dynamics could be specially enhanced by the mesoscale eastward transport of essential

chemical supplies for phytoplankton development. If this mesoscale eastward transport is revealed to be quasi-permanent, it could be associated with recurrent bloom events in this area but this assumption requires further analyses to be generalized.

### 3.3 Fine scale distribution of tracers

The surface tracers' distribution is mostly driven by the wind-induced circulation but also by the transport through mesoscale activity. In a more ephemeral way, tracers can be dispersed following small-scale perturbations such as frontal features. Here

we aim to detect and quantify the influence of such features on the density and chlorophyll surface gradients using both *in situ* and satellite observations.

As described in Section 2.2 and shown by Figures 6a and 6b, the comparison between *in situ* observations and satellite-derived data results in reasonable correlations. The differences between in situ temperature (top), chlorophyll (bottom) and satellite data are plotted on Figure A4. We obtain differences between +1.5°C and -1°C which allows us to confidently use

satellite temperature. For chlorophyll, the differences are smaller than $\pm 0.1$ mg m$^{-3}$ which is also a reasonable deviation between satellite and *in situ* measurements, besides considering the colorbar scale of Figure 6 with values that vary from 0 to 1 mg m$^{-3}$. We can also note that the satellite data clearly underestimate chlorophyll concentrations in the Melanesian area. As both datasets are comparable, SST and chlorophyll concentrations from remote sensing are then used to investigate horizontal

gradients in the WTSP (Fig. 6a and Fig. 6b). To assess the spatial scale of submesoscale gradients (typical of 1-10 km) in terms of ocean dynamics, we use a lagrangian methodology based on the calculation of FSLE (see Sec. 2.3.2) that allow for fronts detection. Figure 6c shows regions where fronts are frequently generated during the time period of the cruise. We notice that the gyre is a region less suitable for fronts to occur and persist. We also identify east of the Fiji islands a zonal band at 18°S where almost no fronts occur during the OUTPACE cruise. Southeast of LDB a mesoscale structure, that is also identified on

Figure 4, creates a frontal barrier that lasted for more than 30 days. It seems that this structure matches with strong surface gradients in chlorophyll and in SST, consequently separating colder and relatively chlorophyll-rich waters to the south from warmer but chlorophyll-poor waters to the north of the front. Overall, as shown by Figure 6, the most frequent and long-lived fronts seem to help in structuring the spatial distribution of tracers such as SST and chlorophyll concentration by creating physical barriers, isolating areas with different biogeochemical characteristics. To try to quantify the influence of frontogenesis

on the structuring effect of surface tracers, we decided to compare the surface sharp gradients measured by the TSG or with the underway fluorimeter with the presence of a front detected from the total surface satellite product.

The strong surface density gradients (as defined in Sec. 2.3.2) represent 9% of the data measured by the TSG during the OUTPACE cruise. The comparison with FSLE reveals that 25% of the strong surface density gradients match with a physical front. Through a bootstrapping re-sampling, the same method is applied to the 91% of TSG data identified as non-gradient,

and demonstrates that only $14 \pm 1\%$ of homogeneous density areas match with a front. These latter results also exhibit that an FSLE existence does not necessarily create a gradient but probably needs pre-existing tracer gradients and a lifetime longer than few days. The same calculations have also been performed for temperature and salinity gradients independently and show similar results. The relatively better correlation between density gradients and FSLEs than between no-density gradients and FSLEs, attests that gradients are not randomly distributed with regard to FSLE structure and proves that FSLE detection can be

a good candidate to explain the presence of *in situ* surface gradients. The same approach was performed with a reactive tracer, the chlorophyll concentration sampled with high frequency. It shows that 35% of strong surface chlorophyll gradients, representative of 1% of the entire underway sampling, agree with the presence of FSLE. Re-sampling over the 99% of non-gradient areas, indicates that $28 \pm 14\%$ of homogeneous chlorophyll areas match with an FSLE.

The correlations with FSLE are not high enough to clearly demonstrate that physical fronts structure the entire surface distribution of tracers. However they give a relative confidence on the fact that they can structure the surface tracers' distribution. The relative orientation of the front with respect to the direction of the density gradient can also be a factor that controls the generation of a strong gradient. The zonal characteristic of the OUTPACE section forces the surface gradient identified with the

TSG to be mainly representative of cross track fronts. Moreover the lack of precision of the calculation method may also cause

a decrease in the correlations calculated. Despite the effort to increase the altimetry resolution to 1/8°, it is still not enough to fully resolve the submesoscale gradients. Consequently the correlation between surface gradients and FSLE can probably not increase much higher than the 25% calculated here. This result converges with Hernández-Carrasco et al. (2011) who showed that FSLEs can still give an accurate picture of Lagrangian small-scale features despite some missing dynamics. Additionally,

our method relies on a comparison between absolute values of gradients and co-located FSLE. Due to the lack of resolution of satellite products, this point by point comparison may induce a few kilometer offset between the area identified with FSLE and the gradient sampled with the TSG. Consequently, the method applied here may not identify the match. Therefore the methodology could be improved by focusing on FSLE values around a certain radius from the position of the gradient to eliminate the uncertainty caused by the absolute point by point comparison. Another way to improve the method would be to

only take into account cross-front gradients that are more likely to be induced by a physical front than along-front gradients. However, considering that the SST distribution is also governed by other processes than advection, such as the diurnal cycle for example, the 25% correlation is large enough to sustain the idea that the submesoscale circulation can participate actively in the spatial structuring of surface tracers such as SSS, SST or density. Concerning the biological chlorophyll parameter, the high percentage (28 %) of FSLEs matching with a «no chlorophyll-gradient» area gives little confidence on the fact that 35%

of chlorophyll gradients were actually caused by the presence of a physical barrier. As chlorophyll concentration is driven by many biological processes, it may be more accurate to associate gradients of phytoplankton abundances, responsible for chlorophyll gradients, with small-scale features. Hereinafter, using two case studies of plankton high frequency sampling, we propose to compare microbial abundances with frontal features.

### 3.4 Example of physical barriers' influences on phytoplankton community

In this section we present two case studies that highlight the potential influence of fronts on phytoplankton horizontal distribution. To test the hypothesis of Bonnet et al. (2015), that pointed out the use of FSLE to explain some correlations between *Trichodesmium spp.* abundances and gradients, we measured the abundances of microbial groups of plankton in samples collected during two high-frequency sampling transects (Fig. 7 and 8). LDB high frequency sampling crosses from North to South the bloom patch described in de Verneil et al. (2017b). The spatial distribution of organisms presents relatively high concentra-

tion of bacteria at the center of the bloom but decreases when exiting this feature (Fig. 7). An FSLE barrier is visible near the center of the transect. This barrier coincides with what is identified as a strong density gradient by our methodology, depicted in Section 3.3. It is also associated with a sharp decrease in surface chlorophyll concentration (Fig. 7b). The variations in the abundance of *Synechococcus*, HNA and LNA (bacteria in general) follow the pattern of surface chlorophyll. We can also notice a slight decrease of PPE abundance to the south of the front. *Prochlorococcus* show different variations: the abundance seem to

increase when crossing the front and then decrease to the south of it. There is a sharp increase of TDIP when exiting the patch in the south, indicating that phosphorus is quickly consumed inside the bloom. The front thus seems to create a barrier for certain organisms which grow and accumulate on one side of the front as demonstrated by relatively high surface chlorophyll (peak at 0.8 $\mu$g l$^{-1}$) and low phosphorus. For LDA high-frequency sampling (Fig. 8a), we can notice a region of high abundance of bacteria at 165°E 15' bounded by two FSLE barriers. This trend is confirmed by figure 8b which shows a relative increase of

the abundance of *Prochlorococcus*, bacteria, HNA and LNA associated with a spike of FSLE values at 45 km. Interestingly, the abundance of PPE seems to decrease where bacterial abundances are the highest indicating that condition favouring bacteria and picocyanobacteria are not necessarily favourable to PPE. In contrast, another FSLE peak at 75 km was characterized by the decrease of *Prochlorococcus*, bacteria, HNA and LNA while the PPE abundance increases. The surface chlorophyll fol-
lows the same pattern as that of *Prochlorococcus*, bacteria, HNA and LNA with a relative increase of chlorophyll concentration ( 0.3 $\mu$g l$^{-1}$) within the region bounded by the FSLEs. On the contrary, *Synechococcus* abundance does not show any variations that coincide with submesoscale features. Another peak in chlorophyll concentration is noticeable around 30 km ( 0.4 $\mu$g l$^{-1}$) but may be associated with other organisms than those analysed with cytometry.

The previous observations suggest that physical barriers to transport, detected with FSLE, can influence phytoplankton community structure by separating or concentrating different picoplankton group. In the LDB case study, the front seems to act like a barrier along which some picoplankton groups aggregate with weak possibilities to cross the front. As a consequence, bacteria and phytoplankton grow on one side of the front whereas, on the other side, the abundances are quite low. According
to Mann and Lazier (2013) phytoplankton growth may be stimulated in aggregates where they can easily take up nutrients released by bacterial decomposition of organic matter. This phenomenon may explain why the abundance is more important on one side of the front. It may also support the persistency of the bloom during LDB. Indeed the bloom may be sustained in time by submesoscale features creating an aggregation of microorganisms that can benefit from each other. de Verneil et al. (2017b) identified the influence of the mesoscale horizontal processes in generating the bloom. They also showed that the bloom was
bounded by some physical features acting like barriers. The results previously presented in this paper support the significance of horizontal submesoscale processes in driving the bloom's dynamic. Here we show that the distribution of phytoplanktonic community inside the bloom is conditioned by submesoscale features.
Our case study at LDB indicates that submesoscale features could create favourable conditions for certain plankton groups at the expense of others. A similar phenomenon occurs at LDA: the abundances of PPE decrease inside the region bounded
by FSLEs where bacterial abundances increase. This observation probably indicates that PPE may not find an advantage in these features. Thus the physical fronts not only structure the spatial distribution of organisms by creating barriers but seem to create border regions influencing the community structure, abundances and diversity. Indeed modelled fine-scale structures have already been shown to delimit niches of different phytoplankton types but also to modify phytoplankton assemblages and diversity (d'Ovidio et al., 2010; Lévy et al., 2014a). Consistent with these previous numerical results, our *in situ* observations
show that microbial growth may benefit from the horizontal conditions engendered by frontal structures. Marrec et al. (2017) also recently observed a peculiar distribution of phytoplankton types inside and outside a mesoscale structure, demonstrating the structuring effect of meso- and submesoscale dynamics on phytoplankton communities. It is also interesting to note that, for our case studies, some picoplankton groups horizontal distribution is not necessarily impacted by the presence of submesoscale features, as for *Synechococcus* during LDA or respond with a different dynamic as for *Prochlorococcus* during LDB. This
observation demonstrates that physical features are a key component but not the only parameter that drives the horizontal

variations of phytoplankton communities. Indeed some patterns can also be explained by inherent biological dynamics. In the context of the OUTPACE cruise, it thus remains important to investigate the role of $N_2$-fixing organisms during these two case studies and to determine how the organisms implied in $N_2$ fixation respond to the presence of submesoscale features (Bonnet et al., 2015).

## 4    Conclusions

We document here the surface circulation at different spatial scales (from 1000 km to 10 km) and its influence on horizontal dispersal of biogeochemical components in the WTSP during the OUTPACE cruise. This study is conducted thanks to the combined use of value-added high-resolution altimetry products, *in situ* observations and Lagrangian numerical simulations. The total altimetry-derived velocity field, combining geostrophy and wind components, revealed a wind-driven meridional pathway of surface waters in the WTSP. This surface trajectory can be linked to the biogeochemical differences between Melanesian waters and gyre waters: a part of the water masses directly recirculates into the gyre whereas the other part is driven across the multiple islands of the WTSP.

The mesoscale activity is confirmed to be intense and mostly westward. Most of these mesoscale structures demonstrated ability to trap waters, however no obvious biogeochemical variations were linked to eddy entrapment. We identify two areas where the mesoscale circulation might have a strong influence on water mass transport. First, the central Coral Sea appears as a region of exchange between distinct NVJ and NCJ waters through eddy transport. Second, the band between 180°W and 170°W could emerge as a recurrent bloom formation area due to the simultaneous effect of $N_2$ fixation, well-known to sustain summertime blooms in the WTSP, and the eastward mesoscale transport of island-enriched waters.

Associating the surface small-scale gradients with Lagrangian diagnostics of frontal features, we showed a correlation of at least 25% highlighting the role of submesoscale activity in governing the horizontal dispersal of surface tracers. The small-scale features also participated in the horizontal distribution and community structure of phytoplankton patches sampled during two original high-frequency sampling of surface phytoplankton abundances.

Future studies in the area will need to take into account the interactions between physical features of the flow at large and fine scales to better understand the phenomenon that drives the distribution of buoyant matter. In particular, the region around station LDB should be investigated during other bloom events to confirm the possible role of enriched-water mesoscale transport in instigating/driving the bloom. This study also revealed the necessity to perform high-frequency sampling during oceanographic cruises to fully resolve submesoscale impacts on biogeochemical distributions.

*Acknowledgements.* This is a contribution of the OUTPACE (Oligotrophy from Ultra-oligoTrophy PACific Experiment) project (Moutin and Bonnet, 2015) funded by the French national research agency (ANR-14-CE01-0007-01), the LEFE-CyBER program (CNRS-INSU), the GOPS program (IRD) and CNES (BC T23, ZBC 4500048836). Solange Duhamel was funded by the National Science Foundation (OCE-1434916). The authors thank the crew of the RV L'Atalante for outstanding shipboard operations, Gilles Rougier, Dominique Lefevre and Francesco d'Ovidio for their valuable help on different dataset. We also thank C. Dupouy for providing high-frequency chlorophyll

concentrations. This work is supported by CLS in the framework of CNES funding and the authors would like to thank M-I Pujol and G. Taburet for provinding enhanced satellite data.

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

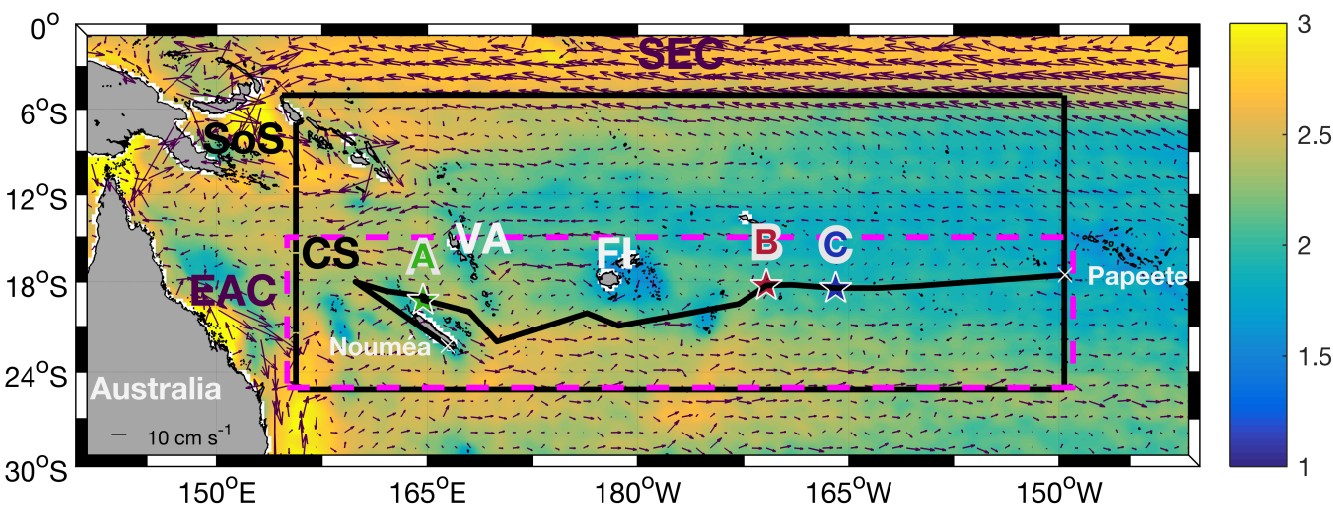

**Figure 1.** Mean eddy kinetic energy (log scale, colorbar) with surface velocity (arrows in cm s$^{-1}$) computed for 10 years (2005 - 2015) from total altimetry-derived surface velocity field. The South Equatorial Current (SEC) and the western boundary East Australian Current (EAC) are indicated. The black line shows the ship track during OUTPACE from Nouméa to Papeete. The positions of the three Long-Duration (LD) stations are drawn with green, red and blue stars for LD-A, LD-B and LD-C respectively. SoS: Salomon Sea; CS: Coral Sea; VA: Vanuatu, FI: Fiji.

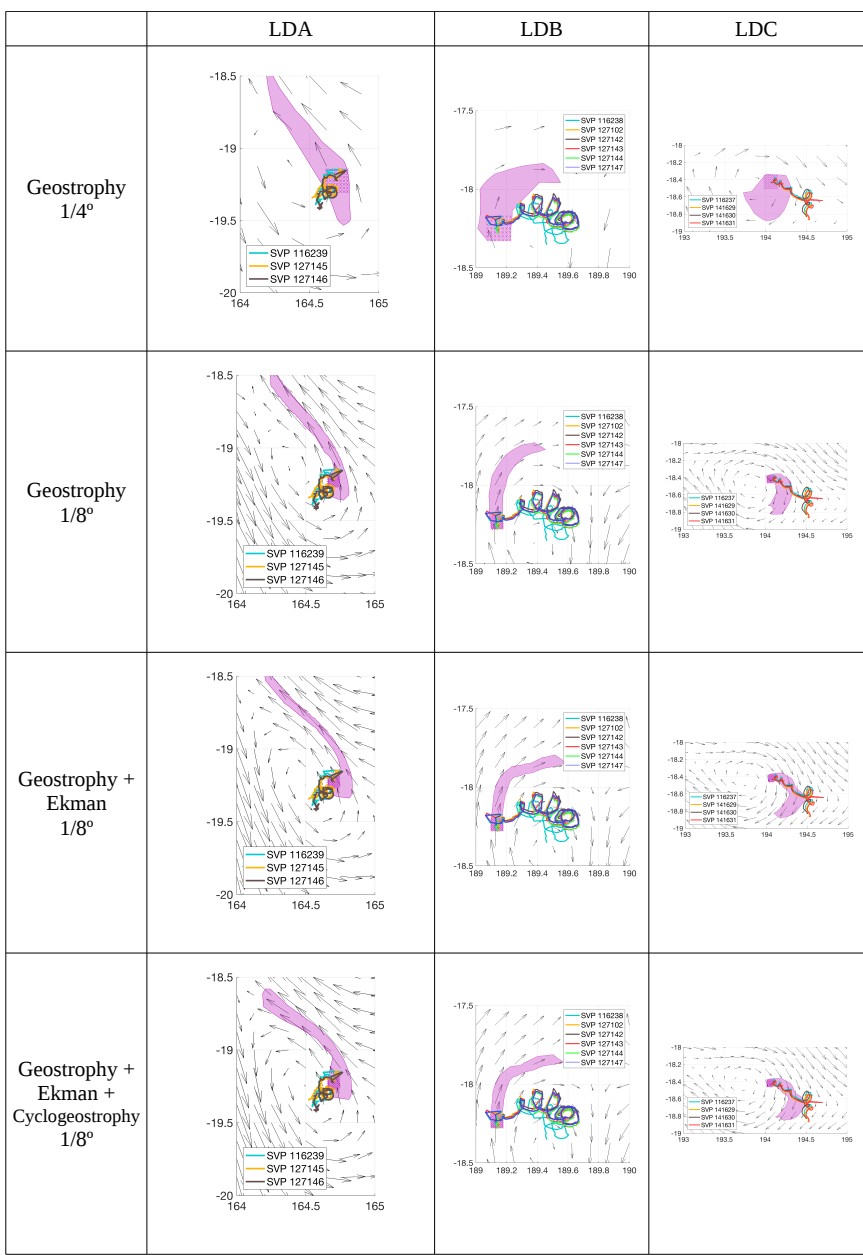

**Figure 2.** Trajectories of numerical particles computed with Ariane (purple patch) with each satellite products and in situ floats (colors) for 8 days after the starting date of each long-duration station. The surface velocity fields of the last day of particle integration (LDA: March, 4; LDB: March, 23; LDC: March, 31) are also shown with black quivers.

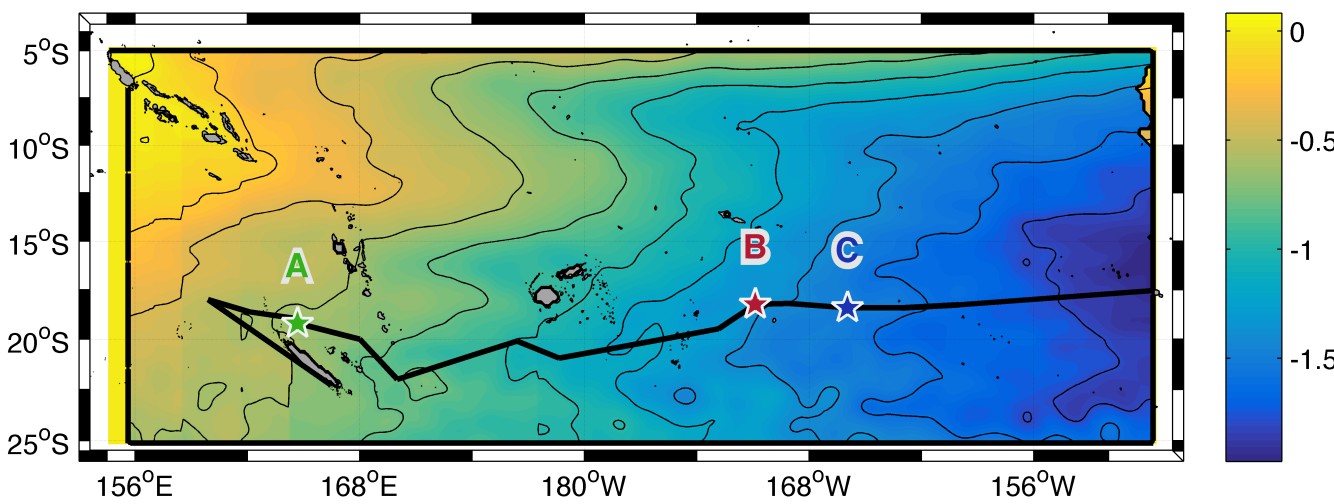

**Figure 3.** Total transport (Sv, colorbar) and streamfunction (black lines with contour interval of 0.25 Sv) computed for ten years with the total altimetry-derived surface velocity field. The ship track and locations of OUTPACE LD stations are indicated with the black line and colored stars as in Figure 1.

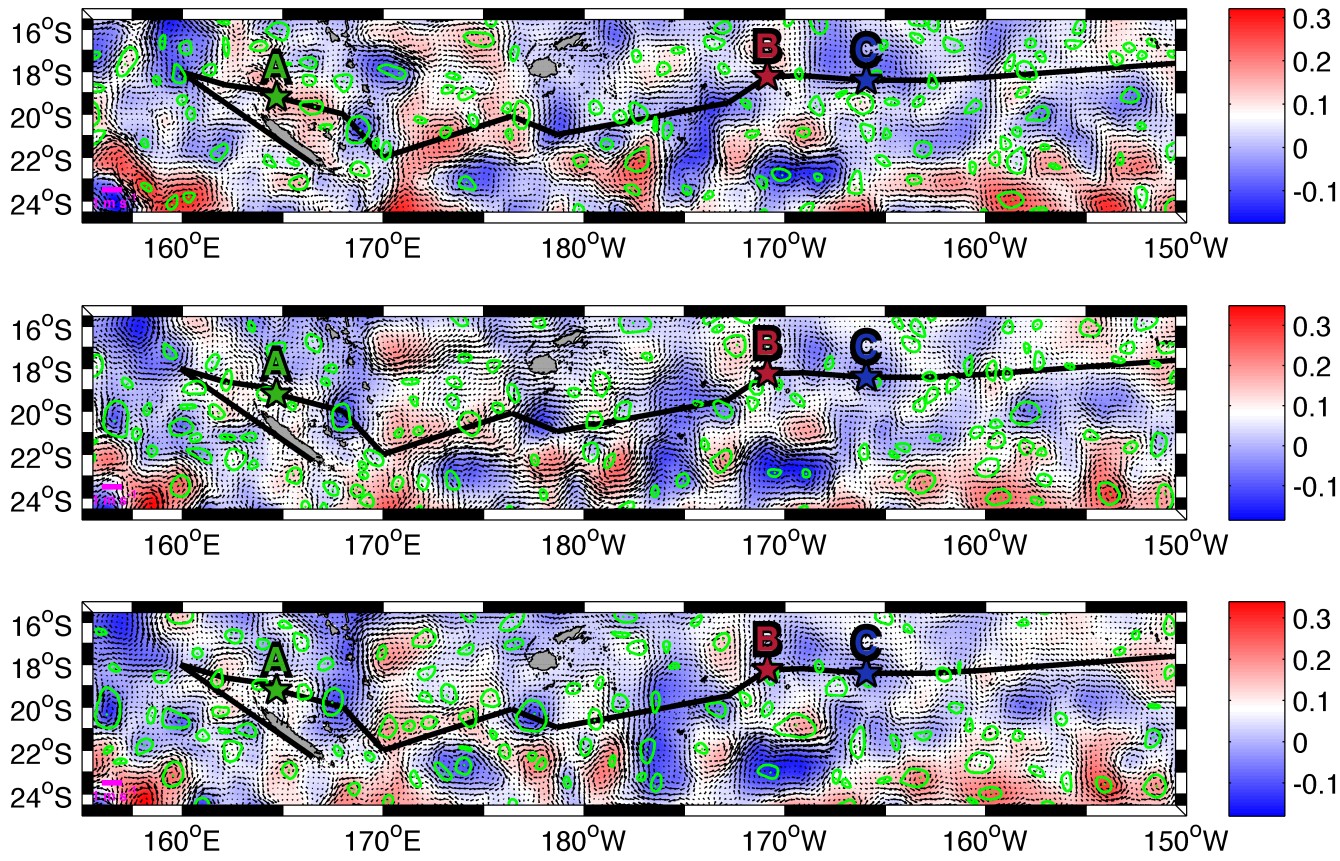

**Figure 4.** Daily sea surface level anomaly (m, colorbar) and velocity field (m s$^{-1}$) from the total altimetry-derived surface product for the first day of LDA (February 25, top), LDB (March 15, center) and LDC (March 23, bottom). Contours of LAVD detected structures are drawn in green. The center of the structures is also indicated by a green point. The ship track and locations of OUTPACE LD stations are indicated with the black line and colored stars as in Figure 1.

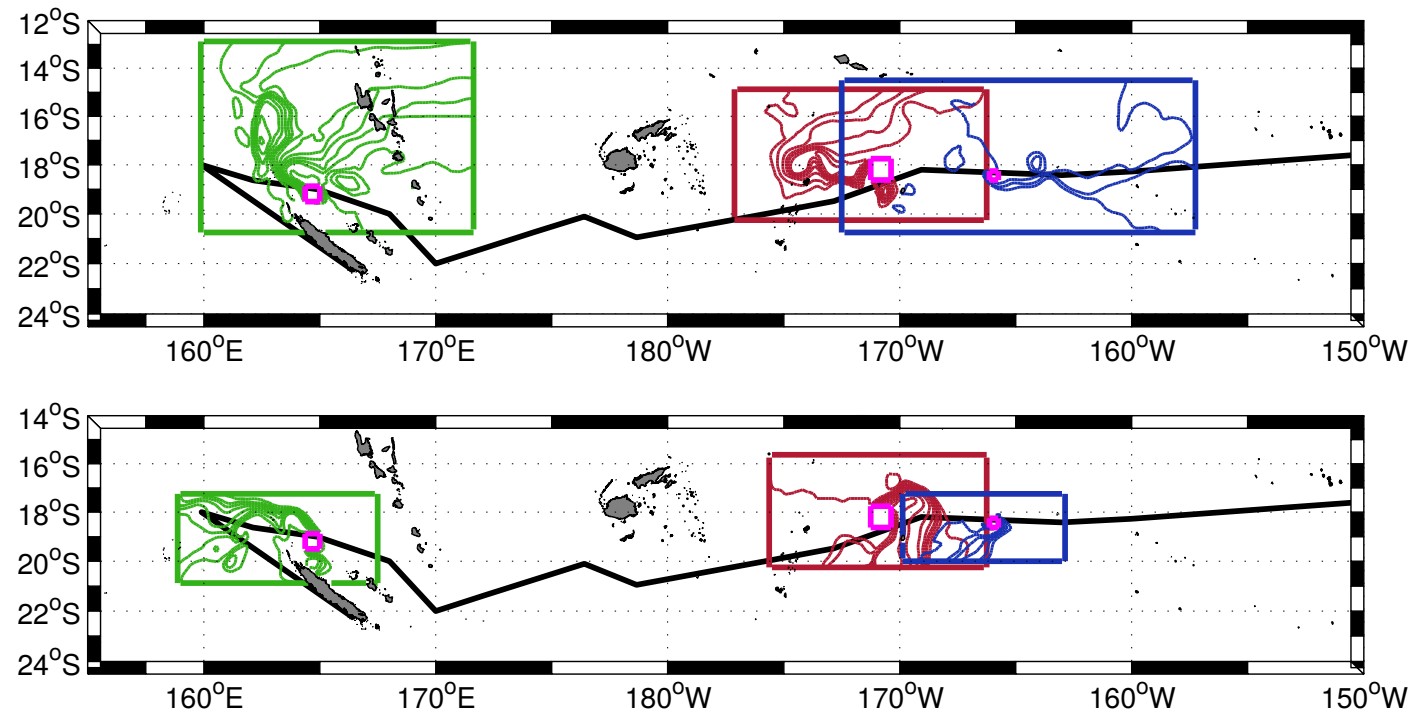

**Figure 5.** Backward (top) and forward (bottom) streamfunctions for LDA (green lines), LDB (red lines) and LDC (blue lines). Numerical particles are initially launched on the magenta boxes which represent the position of each LD station. The domain limit of each Ariane Lagrangian analysis are shown by the large green, red and blue boxes respectively. The ship track of OUTPACE is indicated with the black line.

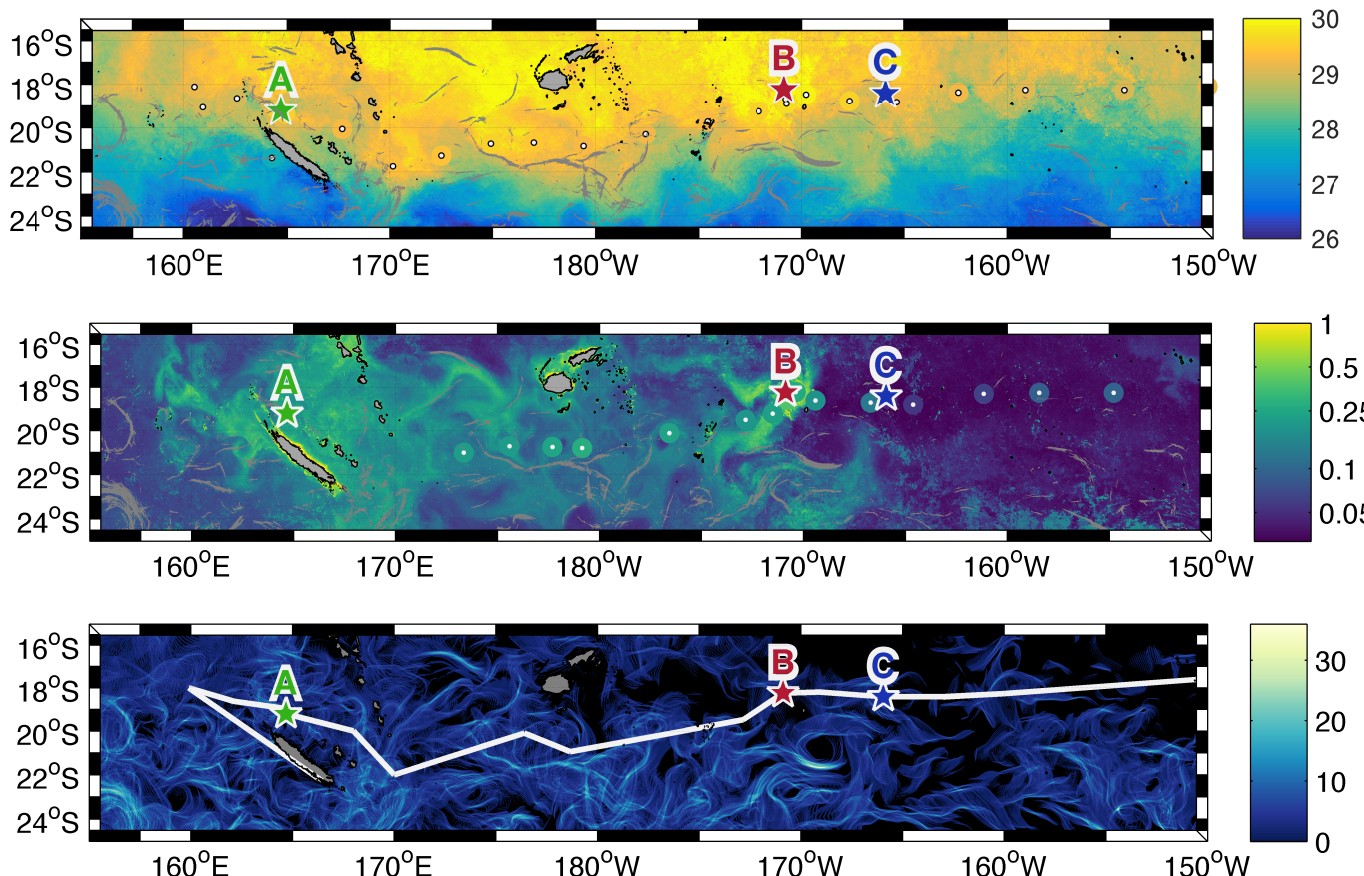

**Figure 6.** Top: Lagrangian satellite sea surface temperature (°C) (for the time period of the OUTPACE cruise, adapted from de Verneil et al. (2017b)) from CLS superimposed with 5 days weighted mean of sea surface temperature (°C) from TSG (colored circles with centres indicated in white). Center: Lagrangian satellite-derived surface chlorophyll concentration (mg m$^{-3}$) (for the time period of the OUTPACE cruise, adapted from de Verneil et al. (2017b)) from CLS superimposed with 5 days weighted mean of surface chlorophyll concentration (mg m$^{-3}$) measured onboard during OUTPACE (colored circles with centres indicated in white). Fronts present for at least 10 days during the OUTPACE cruise are indicated in gray in both figures. Bottom: Recurrence of FSLE structures (number of days of FSLE presence, colorbar).

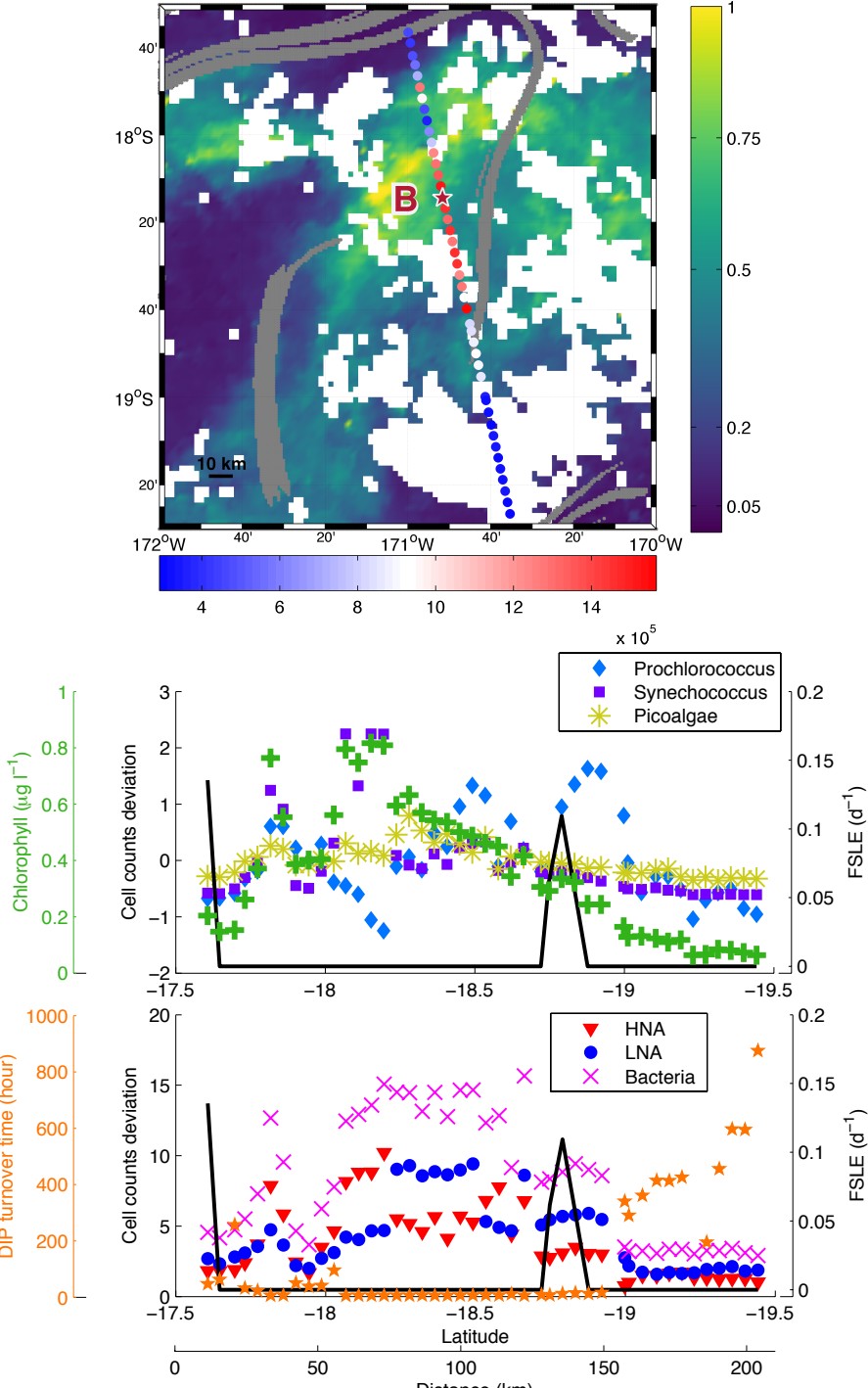

**Figure 7.** Top panel: chlorophyll concentration (mg m$^{-3}$, colorbar) from March 13, 2015 superimposed with bacteria counts (cell mL$^{-1}$, red-blue colorbar) sampled during LDB. FSLE fronts (values > 0.05 day$^{-1}$) are shown in gray. The location of LDB is indicated with the red star. White squares are missing satellite data due to cloud cover. Bottom panel: Cell counts deviation (see Sec. 2.1) of *Prochlorococcus*, *Synechococcus*, PPE and bacteria (HNA and LNA) superimposed with FSLE (day$^{-1}$, black line), surface chlorophyll concentration ($\mu$g L$^{-1}$, green crosses) and dissolved inorganic phosphate turnover time (hours, orange stars) along the high frequency transect of LDB.

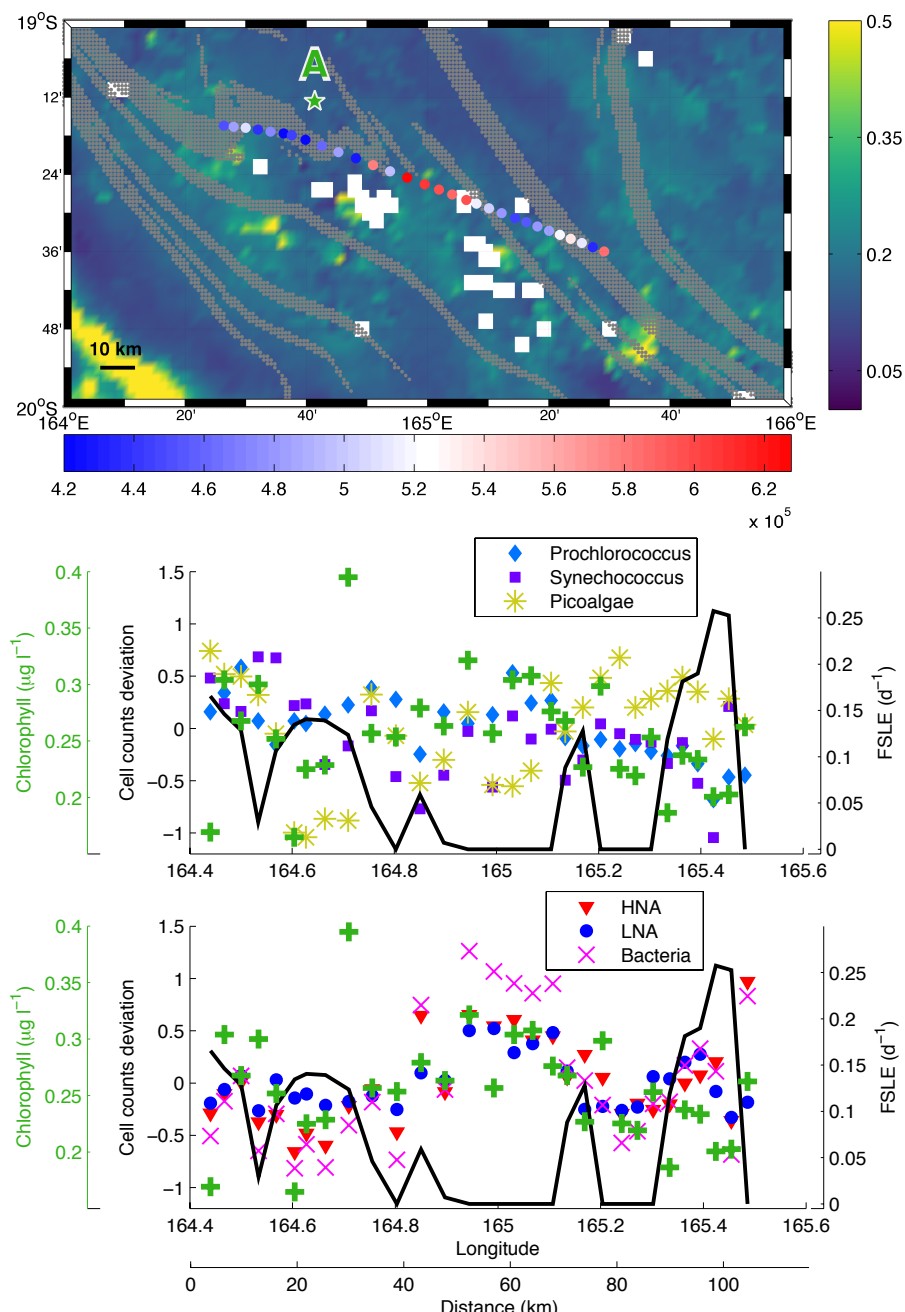

**Figure 8.** Top panel: chlorophyll concentration (mg m$^{-3}$, colorbar) from March 3, 2015 superimposed with bacteria counts (cell mL$^{-1}$, red-blue colorbar) sampled during LDA. FSLE fronts (values > 0.05 day$^{-1}$) are shown in gray. The location of LDA is indicated with the green star. White squares are missing satellite data due to cloud cover. Bottom panel: Cell counts deviation (see Sec. 2.1) of *Prochlorococcus*, *Synechococcus*, PPE and bacteria (HNA and LNA) superimposed with FSLE values (day$^{-1}$, black line) and surface chlorophyll concentration ($\mu$g L$^{-1}$, green crosses) along the high frequency transect of LDA.

**Table A1.** Percentage of particles that return to the initial box (meanders) and percentage of particles that do not reach any controlled sections during the integration (lost) for each backward and forward lagrangian experiments around LD stations.

| STATION | LDA | | LDB | | LDC | |
|---|---|---|---|---|---|---|
| | Backward | Forward | Backward | Forward | Backward | Forward |
| Meanders | 39% | 26% | 65% | 18% | 70% | 44% |
| Particle lost | 22% | 34% | 2% | 18% | 3% | 2% |

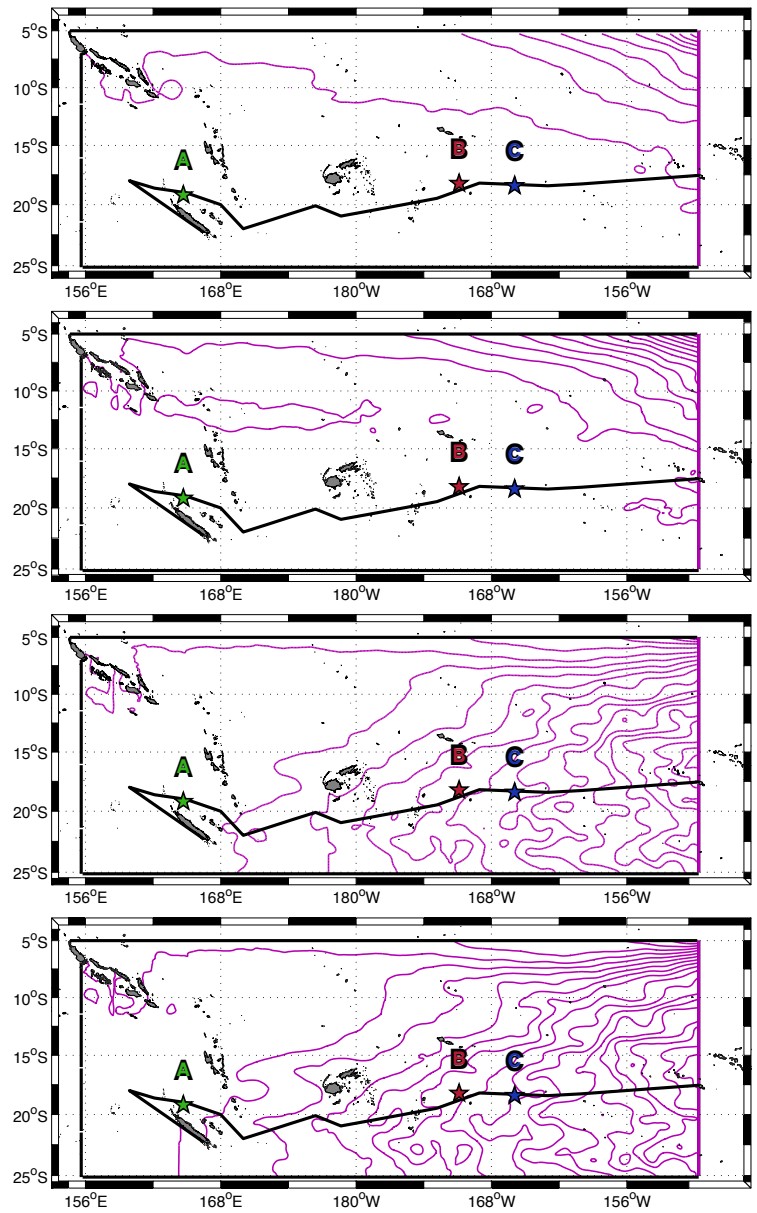

**Figure A1.** Forward streamfunctions computed for a ten year period with, from top to bottom: the low resolution geostrophic product of AVISO; the high resolution geostrophic product from CLS; the high resolution geostrophic and Ekman (at 15 m) product from CLS; and the high resolution geostrophic, Ekman (at 15 m) and cyclogeostrophic product from CLS (referred as the total altimetry-derived product in the text). The initial section of Ariane Lagrangian analysis is indicated with a purple vertical line to the east. The ship track and locations of OUTPACE LD stations are indicated with the black line and colored stars as referred to in Figure 1.

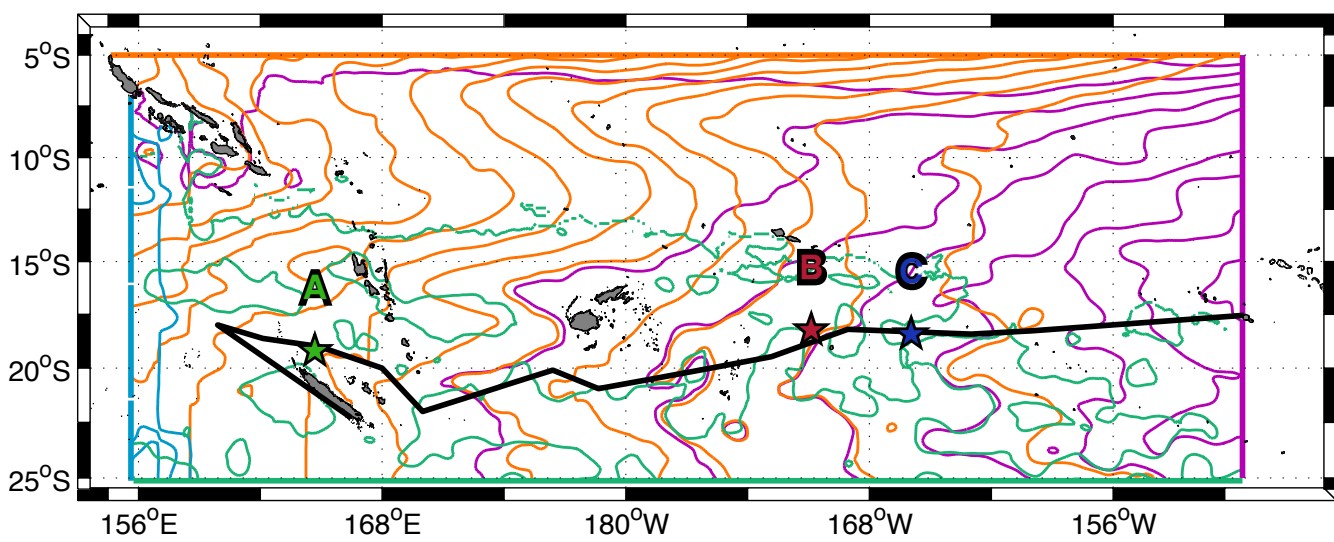

**Figure A2.** Forward streamfunctions computed for a ten year period with the total altimetry-derived product from CLS. Each streamline's color corresponds to the initial section of numerical particles: North section (orange lines), East section (purple lines), South section (green lines) and West section (magenta lines). The ship track and locations of OUTPACE LD stations are indicated with the black line and colored stars as in Figure 1.

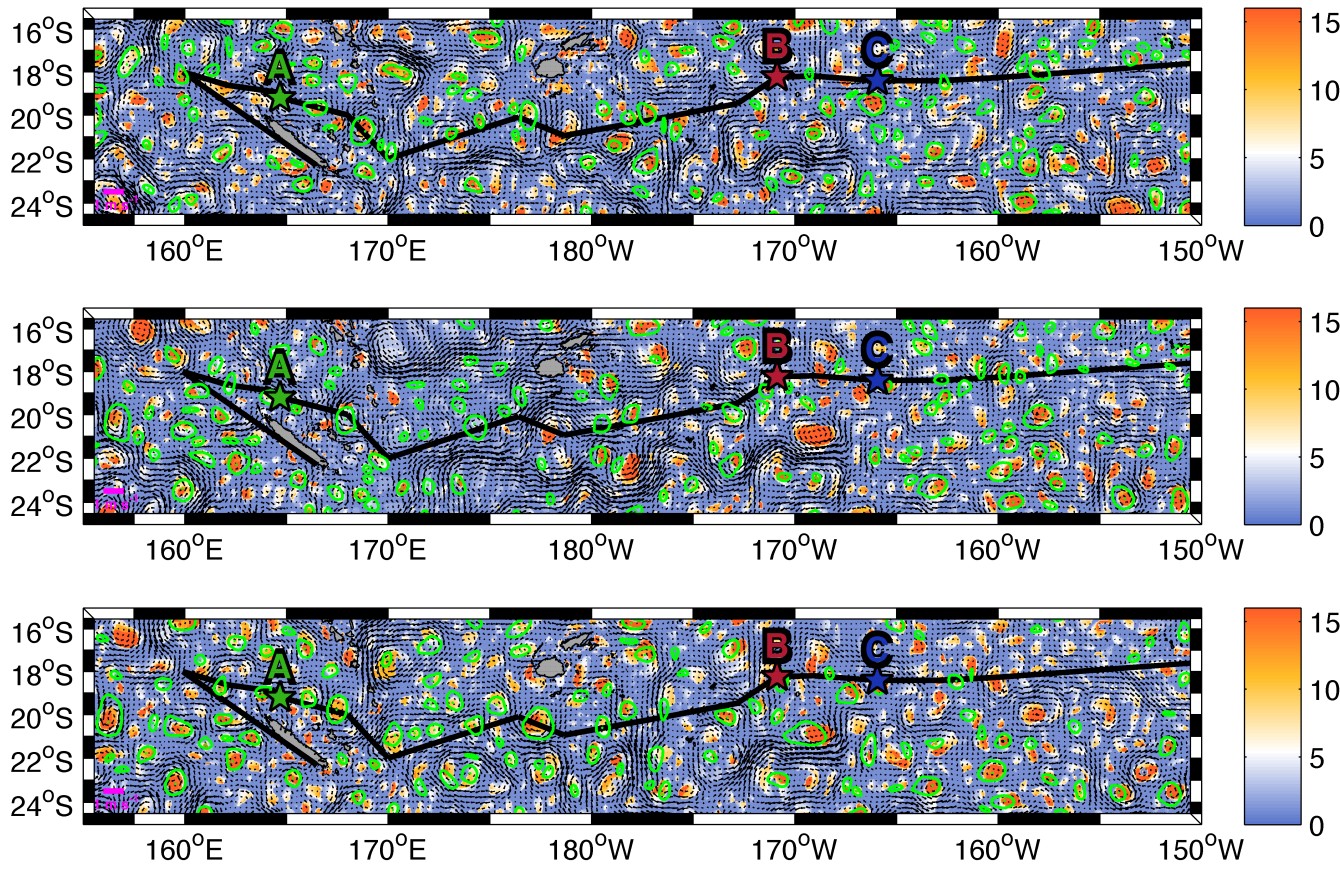

**Figure A3.** Particle retention time (days, colorbar) and velocity field (m s$^{-1}$) derived from the geostrophy, Ekman and cyclogeostrophy included product for the first day of LDA (February 25, top), LDB (March 15, center) and LDC (March 23, bottom). Contours of LAVD detected structures are drawn in green. The center of the each structure is marked with a green point. The ship track and locations of OUTPACE LD stations are indicated with the black line and colored stars as in Figure 1.

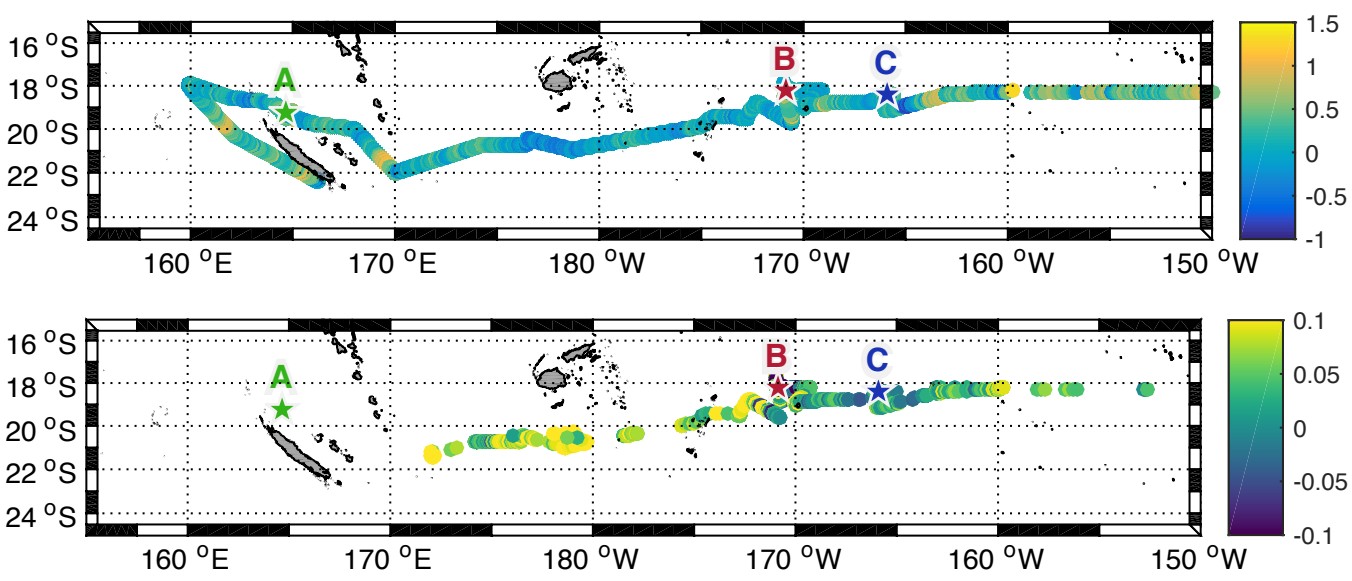

**Figure A4.** Top : Difference between in situ surface temperature from TSG and satellite-derived sea surface temperature from CLS ($^{\circ}$C). Bottom : Difference between in situ surface chlorophyll concentration from the underway survey and satellite-derived sea surface chlorophyll concentration from CLS (mg m$^{-3}$).