# Peer review of "Large to submesoscale surface circulation and its implications on biogeochemical/biological horizontal distributions during the OUTPACE cruise (SouthWest Pacific)"

_Biogeosciences, 2017_

## Referee Comment (RC1) · Anonymous Referee #1 · 4 Dec 2017

In seeking to explore causes and consequences of spatial gradients in the Western Tropical South Pacific, as part of OUTPACE, this manuscript tackles the role of physical advection at different scales. More specifically it investigates how the circulation at large, mesoscale and submesoscale relates to the observed biogeochemical and biological fields. This is important context for the OUTPACE experiment and should be published. There are issues I'd like to see addressed before I can recommend this.

The most significant issue is that the authors overstate the robustness of their results. It would always have been a difficult task to interpret the influence of the physical circula-

tion at all scales studied given the linear nature of the cruise. The claim of demonstrating 'the influence of fronts in controlling the distribution of bacteria and phytoplankton' on the basis of 2 transects and a weak (but admittedly significant) correlation is optimistic. An example of this is the interpretation of Fig 6 on page 12 (lines 21-23). The Chl and other fields decrease over a much larger area than the location of the "tip". The "tip" could presumably have occurred anywhere along this section of decreasing biological fields with the authors drawing the same conclusion.

The combination of Results and Discussion risks being misleading as in several places statements based on direct observation are followed by conjecture written in a similar direct way, sometimes with neither data nor further analysis nor reference to support them e.g. discussion of El Nino and winds at end of Section 3.1 ("data not shown"), "eddy-eddy interactions might be responsible for the emergence of complex paths" on page 10 (line 11), lines 13-18 on page 10, talking of microbial growth with no observations of it on p13 line 9, "isolating areas with different biogeochemical characteristics" on line 17 on page 11. The latter in particular is over-played. Figures 6 and 7 are interpreted as showing coincidence of FSLE and organisms or segregating organisms but this might be guided by the eye of the faithful. Interpreting a "relatively better correlation" (page 11, line 27) as evidence for "not randomly distributed" with 75% of cases still not showing a match is another example as the upper threshold of 25% that is possible from satellite altimetry comes with no evidence to support it. I don't have a problem with conjecture as I think it's an important means of directing future research, but I would recommend pushing analysis further to back these thoughts up and either splitting Results and Discussion or else making much clearer where the reporting of observations ends and speculation begins.

The comparison of satellite-based advection proxies with drifter data seems to be a more significant piece of work than acknowledged here and I would like to see it rescued from the Appendix and given a more thorough account in the main body of the paper. As part of this it would be good to see a discussion of the possible bias of
comparing trajectories of just a few real floats with many more virtual ones and an acknowledgement of the fact that the streamfunctions only really do a reasonable job compared to drifter trajectories for LDC (Figure A1) and why this might be so.

Minor:

- Page 3, line 6: I'm not sure I'd describe the surveyed area as 'relatively high kinetic energy' or 'intense' given that it only visits the high value areas intermittently in Fig. 1.

- Page 4, line 21: how was DIP turnover time calculated and why is it of interest?

- Page 7, line 33: "...and 0.2...as thresholds..."

- In several places 'west' and 'east' are confused. e.g. page 8, line 27; page 10, line 2; page 10, line 28

- Page 8, line 29: explain location/extent of Melanesia

- Page 11, line 7: I think it is debatable that chlorophyll shows a "reasonable correlation in Fig. 5. Scatter plots and correlation n coefficients would be more convincing.

- Figure 4: The backward and forward streamfunctions cross, particularly for LDA. This warrants comment.

- Page 22, Fig 5 middle: why are the data points for the TSG so far apart given that it is a flow-through system?

- Figures 6 and 7: it is difficult to relate top and bottom panels with one labelled in degrees and the other in km.

- Figure 7: what are the white squares? Missing data due to cloud?

---

## Referee Comment (RC2) · Anonymous Referee #2 · 14 Dec 2017

Review of "Large to submesoscale surface circulation and its implication on biogeo-chemical/biological horizontal distribution during the OUTPACE cruise" by L. Rousselet et al.

This paper aims to highlight the role of large to submesoscale surface circulation on biogeochemical/biological horizontal distribution during the OUTPACE cruise. This cruise, held in Feb-April 2015, was dedicated to obtain a precise representation of the complex interactions between planktonic organisms and the cycle of biogenic elements along a zonal section in the WTSP ocean crossing contrasting environments

(from oligotrophy to ultra-oligotrophy). Results from this cruise are being valorized and some papers are already published. Moutin et al. (2017) described the hydrological and dynamical context of biogeochemical sampling. De Verneil et al. (2017) analyse a surface chlorophyll a bloom by combining in situ and remote sensing datasets. They characterize the role of the physical circulation, and in particular the role of surface mesoscale circulation responsible for the bloom's biogeochemical properties. This new paper wants to put the OUTPACE cruise into a more synoptic view by looking at surface satellite data in order to illustrate the role of large scale as well as meso/submesoscale dynamics on the biogeochemial/biological horizontal distributions of OUTPACE data. This is clearly a worthwhile exercise, and the authors have done a substantial work by compiling different satellite data set, and by using original diagnostics. The paper is well written, easy to read. The description of surface conditions by satellite data confirms most of the existing literature. But at first order, the motivation of this study is to do the link between this synoptic approach and the OUTPACE zonal section. It is not an easy task, and as a consequence most of the results are highly speculative. It is not really discuss what is the new contribution of this paper in comparison with the other existing papers dealing with the OUTPACE data. So, at this stage I am not convinced that the materials presented are strong enough to justify a publication. My general comment is to not recommend the paper for publication until an effort was done to better give evidence of the results that motivate this paper.

In the following I address some remarks whishing that it could help the authors to revise their manuscript.

l.4: "several studies indicate a strong mesoscale variability due to barotropic instabilities and the interactions of the major currents and jets with the numerous islands of the region (Qiu et al., 2009; Hristova et al., 2014)." Barotropic instability doesn't appear as the only source of instability in the WTSP. If barotropic instabilities are related to the NVJ/CSCC/NCJ system (at 16°S) in most places baroclinic instabilities prevail as in the STCC-SEC system (Qiu and Chen, 2004; Hristova et al., 2014).

General comment: A great number of satellite data are used. Four different altimetry-derived velocity product are tested: The classic Ssalto/Duacs product that provides maps of absolute geostrophic velocities at $1/4°$ of resolution, a similar product at higher resolution ($1/8°$) that may include an Ekman component, and a cyclogeostrophy correction. The authors identify the last product as the most accurate in situ surface currents with regards to the trajectories of the SVP floats launched during OUTPACE. It is not surprising than adding the Ekman component improve the comparison with surface drifters. The cyclogeostrophy correction must improve the current associated with eddies by taking account centrifugal acceleration. In my knowledge, it is a really new altimetric product and it could be interesting to illustrate the improvement brought by this correction. On this part, I am reserved because we don't know what are the respective contributions of high resolution, Ekman component, cyclogeostrophy. The result is based by looking at trajectories of numerical particles launched at only the three particular locations where SVP floats were deployed. Besides the relative limited number of locations for the validation, it is difficult to validate such conclusion based only on Fig. A1. If the LDC plot provides a good qualitative agreement, it is not the case for LDA and LDB.

p.6 l.29: "It is clearly more interesting to use CLS data instead of commonly used AVISO geostrophic surface currents because they include the wind effect and cyclogeostrophy with higher resolution, as well as better represents in situ data." Okay, I agree that the last product should represent in a better way the surface circulation. The authors argue that it is the most interesting data set. I am not against this opinion but it depends on the objective. Is it really interesting to add an Ekman component when you are interested by mesoscale features? The Ekman velocity is a strong component of the surface circulation but it is limited to the thin Ekman layer. So what is the importance of transport in such layer with regard to OUTPACE objectives where processes on the vertical must be crucial and partly related to submesoscale features? In my opinion, it lacks a discussion on the interest of such product for your objective.

p.7 l.1: Figure 4 is cited before Fig. 3

Table 1: For me it is mysterious why the statistics of meanders vary so much between backward/forward experiments. Any explanation?

General comment: The authors have used Lagrangian approaches based on the recent LAVD method, and the combination of OW and RP parameters used in d'Ovidio (2013). They are very interesting methodologies as they ensure that eddies correspond with trapped waters. It seems more robust that the more classic linear parameter used in Chelton et al. (2011). As it is written both methods are in good agreement so what is the advantage to mix both methods? There is always some subjectivity when defining eddies, also when looking at Fig. C2, it seems that the RP contours are more in agreement with the velocity field. If one result of this paper is to test different methods to detect and track eddies, it could be interesting to test the methods classically used based on sea level extrema (Chelton, 2011; Chaigneau et al., 2009, Isern Fontanet, ).

FSLE results are compared with surface gradient measured from the in situ OUTPACE data. Because the OUTPACE section is zonal, the corresponding surface gradient is mainly representative of cross track fronts. This aspect that limits the comparison of the two datasets is not mentioned.

p.8 l7: "using altimetric..including the wind effect" and the cyclogeostrophy correction?

P.8 l.14: "or an artefact.. due to the short averaging.." It seems in contradiction with the sentence in section 2.3.1: ".. these simulations ensures that the use of a one year time period doesn't significantly modify statistical outputs". Also, with a RT of 15 days max for eddies it would be surprising that mesoscale dynamics have a strong influence on the mean circulation.

p.8 l.15: "The meridional transport does not correspond to any surface current but is mainly due to the south-easterly trade winds. This meridional component appears due to the addition of the Ekman component in altimetry surface velocities (Fig.B1, Supplementary Material). The large scale transport of surface waters in the OUTPACE area is thus a combination of the transport by general well-known surface currents and wind-driven circulations." What are the dynamics which refer to these "surface currents"?? This sentence is a little bit ambiguous. If the "surface current" refers to geostrophic balance, it is normal that it doesn't take account for the well-known poleward Ekman transport. Now if we look at the circulation inferred from Sverdrup theory, it is not so different of the streamfunction in Fig.2 despite its depth integrated estimation.

P.8 l18: "surface waters travel from northeast to southwest" Fig. C1 shows that most of the waters of importance for OUTPACE comes from the north and the northest as written in l.22.

p.8 L27; "west of 170°w" → East of??

P.8 28: ".The wind-driven surface transport highlighted here could participate in the bio-geochemical variations between western and eastern waters. Indeed the path through the Melanesian area may enrich these waters due to the contact with multiple islands whereas waters that directly recirculate within the gyre keep their ultra-oligotrophic characteristics. " The authors here want to do a link between the wind driven circulation at the surface and biogeochemical variations. It seems very speculative because their argument is the path through the Melanesian area that is also valid for what is named "surface current". Also, what is the role of these surface waters in the biochemical properties against deeper water and vertical processes??

p.9 l.4: "the OUTPACE cruise took place during an El Niño phase but they determined that climatological effects, upon the results of the cruise, were minimized." The trade winds are very sensitive to ENSO conditions in the WTSP. So the authors argue for little effects of the wind driven circulation upon the results of the cruise. So, it seems to be in contradiction with the fact to use the altimetric product including the wind effect. Also, the authors argue the interest to investigate the mesoscale circulation. I am not sure that the altimetric product they used is well suited for such purpose. At first order,

meso and submescale are driven by internal dynamics.

p.9 l. 20: "If the major part propagates westward, the meridional band between 180◦W and 170◦W is identified as a region with mostly eastward propagation of mesoscale structures." Based in Fig.3, it is very hard for me to see propagation. This result is developed in the next sentences to argue for the importance of mesoscale dynamics to transport enriched-fluid into nutrient-poor gyre waters. But this eastern propagation is not really shown and as said by the authors there are no in situ data to illustrate this point. At this stage the discussion is highly speculative.

P.10 l. 25 " enhanced by the mesoscale transport" Fig.4 shows an eastward transport at LDB but the link with mesoscale is not obvious. Are there particles trapped into eddies that propagate eastward until LDB? And what their retention time and their distance travelled?

P.11 l. 7 "correlations" It is not really a correlation here

P.11 l.25; "These latter results also exhibit that an FSLE existence does not necessarily create a gradient but probably needs pre-existing tracer gradients and a lifetime longer 25 than few days." The orientation of the front against the direction of the density gradient could be also an explanation?

Fig.6. It should be better if the axis in Fig.6b is in ° (as in Fig. 6a) despite than in km.

Section 3.4 is the section that best fits with the objective of the paper highlighted in the title. It shows two interesting case studies of interaction between fronts and biogeochemical properties. In my opinion, it is the most interesting part of the paper but it is only 1 page. It is regrettable that such results are not discussed with regard to the results of De Verneil et al. (2017). Do strong density gradients correspond with the FSLEs discussed LDB and LDA?

---

## Author Comment (AC1) · 19 Jan 2018

We would like to thank the Anonymous Referee 1 for his review. In the following we respond to the reviewer's comments and we follow most of his proposition in order to improve our work and its presentation. For clarity, the reviewer's remarks are copied in bold.

**In seeking to explore causes and consequences of spatial gradients in the Western Tropical South Pacific, as part of OUTPACE, this manuscript tackles the role**

**of physical advection at different scales. More specifically it investigates how the circulation at large, mesoscale and submesoscale relates to the observed biogeochemical and biological fields. This is important context for the OUTPACE experiment and should be published. There are issues I'd like to see addressed before I can recommend this. The most significant issue is that the authors overstate the robustness of their results. It would always have been a difficult task to interpret the influence of the physical circulation at all scales studied given the linear nature of the cruise.**

Indeed, we agree definitely with your point but, if it is very difficult to fully explore the influences of the physical circulation considering only data collected during the cruise combined with satellite data, it remains one important step to explore before using models and build other observational strategy. Here we wanted to give the most complete picture as possible, given the available data from the cruise, of the circulation at scales that are known to play a major role and their potential influence on biogeochemical variabilities. This, of course, leads to make some assumptions that are not verified yet in this paper. We agree to pay attention in separating the observations part and the hypotheses made to explain them as well as the potential influences raised by these observations.

**The claim of demonstrating 'the influence of fronts in controlling the distribution of bacteria and phytoplankton' on the basis of 2 transects and a weak (but admittedly significant) correlation is optimistic. An example of this is the interpretation of Fig 6 on page 12 (lines 21-23). The Chl and other fields decrease over a much larger area than the location of the "tip". The "tip" could presumably have occurred anywhere along this section of decreasing biological fields with the authors drawing the same conclusion.**

We agree with the reviewer that the focus should not be on the tip of the FSLE as, also considering the resolution of the satellite data, the "real" physical barrier might be offseted by a few kilometers. What is interesting in this studied case is that the FSLE

delimit a region of relatively high abundance of organisms and a region of relatively low abundance on each side of it. In section 3.4, we describe and discuss what appears to be the clear influence of fronts in the two case studies. These are two examples of the potential influence of fronts on the horizontal phytoplankton distribution. We are not suggesting that every front will lead to a separation of phytoplankton community, but that some of them can. Here we propose to rename section 3.4 to "Example of physical barriers' influences on phytoplankton community" and to add the following sentences to help the readers to clearly understand the conditional form on such influence: "In this section we present two case studies that highlight the potential influence of fronts on phytoplankton horizontal distribution. To test the hypothesis of Bonnet et al..".

**The combination of Results and Discussion risks being misleading as in several places statements based on direct observation are followed by conjecture written in a similar direct way, sometimes with neither data nor further analysis nor reference to support them e.g. discussion of El Nino and winds at end of Section 3.1 ("data not shown"), "eddy-eddy interactions might be responsible for the emergence of complex paths" on page 10 (line 11), lines 13-18 on page 10, talking of microbial growth with no observations of it on p13 line 9, "isolating areas with different biogeochemical characteristics" on line 17 on page 11. The latter in particular is over-played. Figures 6 and 7 are interpreted as showing coincidence of FSLE and organisms or segregating organisms but this might be guided by the eye of the faithful. Interpreting a "relatively better correlation" (page 11, line 27) as evidence for "not randomly distributed" with 75% of cases still not showing a match is another example as the upper threshold of 25% that is possible from satellite altimetry comes with no evidence to support it. I don't have a problem with conjecture as I think it's an important means of directing future research, but I would recommend pushing analysis further to back these thoughts up and either splitting Results and Discussion or else making much clearer where the reporting of observations ends and speculation begins.**

We understand the reviewer's concern about the combination of Results and Discussions together as this question was raised during the article scripting process. In consequence, we decided to describe the circulation from large to submesoscale through a descending approach and it was a difficult task to split the Results and the Discussion into two different sections as the Discussion will refer to different sections of the Results part. To avoid the reader a back and forth exercise between what was described in the Results and what was suggested in the Discussion, we believed (and still do) it is more convenient to directly discuss the results highlighted. So we tried to be extremely rigorous with the tenses: using the direct way to talk about the results and the conditional way (could, may, might...) to talk about the potential influence of the observations. However we understand that some parts, listed by the reviewer, were still suffering from a lack of clarity. To avoid confusion, we propose to clearly split the results and discussions into different paragraph in each existing subsections (3.1, 3.2, 3.3, 3.4) and to add few sentences to clarify the transition between the observations of the circulation and the potential impact on biogeochemical distributions:

- Section 3.1 "Considering the large scale biogeochemical distribution, the meridional transport observed could lead to ..."
- Section 3.2 "Coherent mesoscale features are well-known to participate in the surface biogeochemical variations ..."

**The comparison of satellite-based advection proxies with drifter data seems to be a more significant piece of work than acknowledged here and I would like to see it rescued from the Appendix and given a more thorough account in the main body of the paper. As part of this it would be good to see a discussion of the possible bias of comparing trajectories of just a few real floats with many more virtual ones and an acknowledgement of the fact that the streamfunctions only really do a reasonable job compared to drifter trajectories for LDC (Figure A1) and why this might be so.**

We agree with the reviewer's suggestion and we propose to add a new subsection in

Material and Methods to detail the comparison between the different satellite-based advection and in situ drifters. The new subsection would be *2.4 Comparison of satellite products with in situ drifters*. It would include the Figure 1 and discussion raised in the third comment of Referee#2.

**Minor: Page 3, line 6: I'm not sure I'd describe the surveyed area as 'relatively high kinetic energy' or 'intense' given that it only visits the high value areas intermittently in Fig. 1.**

The "relatively high eddy kinetic energy" refers also to the study by Qiu et al., 2009. That's why we propose to change the sentence as follows : " As displayed in Figure 1, the OUTPACE cruise was conducted in the transition area between a zonal band of relatively high eddy kinetic energy south of 19°S (Qiu et al., 2009) and low eddy kinetic energy to the north. "

**Page 4, line 21: how was DIP turnover time calculated and why is it of interest?**

As describe in Moutin et al. (this issue), Dissolved Inorganic Phosphate (DIP) turnover time represents the ratio between Phosphate natural concentration and Phosphate uptake by planktonic species (Thingstad et al., 1993). It is considered the most reliable measurement of phosphate availability in the upper ocean waters (Moutin et al., 2008). In our region of interest, the phytoplankton growth, and in particular nitrogen fixers, is often limited by Phosphate availability and Phosphate may appear as a key factor controlling carbon production (Van den Broeck et al., 2004). This parameter thus gives an important information on the biological activity. As these clarifications are important, we add them in Material and Methods : "Dissolved inorganic phosphate turnover times (TDIP) were determined using a dual 14C-33P labelling method following Duhamel et al. (2006) and described in Moutin et al. (this issue). As describe in the latter, DIP turnover time represents the ratio between Phosphate natural concentration and Phosphate uptake by planktonic species (Thingstad et al., 1993). It is considered the most reliable measurement of phosphate availability in the upper ocean waters (Moutin et al.,

2008). In the WTSP, the phytoplankton growth is often limited by Phosphate availability. This parameter thus gives an important information on the biological activity in relation to resource availibility : a very short DIP turnover time means rapid utilization of the ambient phosphate present in limiting concentration, whereas a long DIP turnover time represents a slow utilization of the ambient phosphate present in higher concentration."

**Page 7, line 33: "...and 0.2...as thresholds..." In several places 'west' and 'east' are confused. e.g. page 8, line 27; page 10, line 2; page 10, line 28**

We checked and corrected the points highlighted.

**Page 8, line 29: explain location/extent of Melanesia**

We modify the sentence as follows : " Moreover the path through the Melanesian area, which includes the multiple islands from Papua New Guinea to Fiji (140°E-170°W), may enrich these waters due to the contact with islands whereas... "

**Page 11, line 7: I think it is debatable that chlorophyll shows a "reasonable correlation in Fig. 5. Scatter plots and correlation n coefficients would be more convincing.**

Correlation coefficients are mentioned in Section 2.2 p5 L34 : 0.8 for both temperature and chlorophyll, which is a reasonable value when comparing in situ data and satellite-derived data. Below we plot the difference between in situ temperature (chlorophyll) and satellite data (Fig. 1 and 2 respectively). In temperature, we get differences between +1.5°C and -1°C which allows us to confidently use satellite temperature. For chlorophyll, the differences are smaller than $\pm0.1$ mg m$^{-3}$ which is also a reasonable deviation between satellite and in situ measurements, besides considering the color-bar scale of Figure 5 with values that vary from 0 to 1 mg m$^{-3}$. We can also note that the satellite data clearly underestimate chlorophyll concentrations in the Melanesian area. We believe this figure and explanation could help the reader, so we decide to add the figure in the Appendix and to add this text in Section 3.3 : "The differences

[Figure]

between in situ temperature (chlorophyll) and satellite data are plotted on Figure A4 (top and bottom respectively). In temperature, we get differences between +1.5°C and -1°C which allows us to confidently use satellite temperature. For chlorophyll, the differences are smaller than ±0.1 mg/m$^{-3}$ which is also a reasonable deviation between satellite and in situ measurements, besides considering the colorbar scale of Figure 5b with values that vary from 0 to 1 mg/m$^{-3}$. We can also note that the satellite data clearly underestimate chlorophyll concentrations in the Melanesian area."

**Figure 4: The backward and forward streamfunctions cross, particularly for LDA. This warrants comment.**

We agree with the reviewer: this is very interesting that around LDA backward and forward paths cross. This highlights again the complexity of the circulation between New Caledonia and Vanuatu, characterized by meanders and recirculations. We propose to add a comment about this in Section 3.2 as it reinforces the previous observations of complex path in this area (Rousselet et al., 2016) : "Backward and forward streamfunctions cross around station LDA which suggests that the area between New Caledonia and Vanuatu is a region of complex recirculation with waters that stay in this region for a while before exiting the Coral Sea."

**Page 22, Fig 5 middle: why are the data points for the TSG so far apart given that it is a flow-through system?**

Indeed the TSG provides data every 1min30, however plotting these data on a scatter plot including the whole cruise route requires a large amount of memory to finally obtain a not so clear information. Thus to reduce the size of the figure and provide an useful comparison with the satellite image, we decided to plot a weighted mean over 5 days that is the time interval used to produce the composite satellite images. As a consequence the position of the point depends on the position of the boat every 5 days. We performed the same calculation for underway chlorophyll data (Fig. 5b). In order to avoid misunderstanding, we precise in the caption that TSG and chlorophyll data point

correspond to weighted mean data over 5 days as follows : " Top : ... superimposed with 5 days weighted mean of sea surface temperature (°C) from TSG... .Center : ... superimposed with 5 days weighted mean of surface chlorophyll concentration...".

**Figures 6 and 7: it is difficult to relate top and bottom panels with one labelled in degrees and the other in km.**

Referee #2 also pointed out this issue. We agree to change both figures and to plot Fig 6b. and Fig 7b. in degrees (Fig.3 and 4).

**Figure 7: what are the white squares? Missing data due to cloud ?**

Indeed white squares are missing data due to cloud covering as we use satellite data from a specific day. This missing information will be added in the caption for more clarity : " White squares are missing satellite data due to cloud cover."

**References**

Duhamel, S., Zeman, F., and Moutin, T. (2006). A dual-labeling method for the simultaneous measurement of dissolved inorganic carbon and phosphate uptake by marine planktonic species, Limnol. Oceanogr. Meth., 4,416–425.

Moutin, T., Karl, D. M., Duhamel, S., Rimmelin, P., Raimbault, P., Van Mooy, B. A. S., and Claustre, H. (2008). Phosphate availability and the ultimate control of new nitrogen input by nitrogen fixation in the tropical Pacific Ocean, Biogeosciences, 5, 95-109.

Moutin, T., Wagener, T., Caffin, M., Fumenia, A., Gimenez, A., Baklouti, M., Bouruet-Aubertot, P., Pujo-Pay, M., Leblanc, K., Lefèvre, D., Helias Nunige, S., Leblond, N., Grosso, O. and de Verneil, A. (this issue). Nutrient availability and the ultimate control of the biological carbon pump in the Western Tropical South Pacific Ocean. Biogeosciences Discussion.

Rousselet, L., Doglioli, A. M., Maes, C., Blanke, B., Petrenko, A. A. (2016). Impacts of mesoscale activity on the water masses and circulation in the Coral Sea. Journal of

Geophysical Research: Oceans, 121(10), 7277-7289.

Thingstad, T. F., Skjoldal, E. F. and Bohne, R. A. (1993). Phosphorus cycling and algal–bacterial competition in Sandsfjord, western Norway. Mar. Ecol. Prog. Ser. 99: 239–259.

Van Den Broeck, N., Moutin, T., Rodier, M., and Le Bouteiller, A. (2004). Seasonal variations of phosphate availability in the SW Pacific Ocean near New Caledonia, Marine and Ecological Progress Series, 268, 1-12.

[Figure]

[Figure]

**Fig. 1.** Difference between in situ surface temperature from TSG and satellite-derived sea surface temperature from CLS ($^\circ$C)

[Figure]

**Fig. 2.** Difference between in situ surface chlorophyll concentration from the underway survey and satellite-derived sea surface chlorophyll concentration from CLS (mg mËL'³ )

[Figure]

**Fig. 3.** Modified figures 6b

**Fig. 4.** Modified figures 7b

---

## Author Comment (AC2) · 19 Jan 2018

First, we would like to thank the Anonymous Referee 2 for his review. In the following we respond to his comments. We followed most of the remarks addressed and we believe the modifications improve our manuscript.For clarity, the reviewer's remarks are copied in bold.

**This paper aims to highlight the role of large to submesoscale surface circula-tion on biogeochemical/biological horizontal distribution during the OUTPACE**

**cruise. This cruise, held in Feb-April 2015, was dedicated to obtain a precise representation of the complex interactions between planktonic organisms and the cycle of biogenic elements along a zonal section in the WTSP ocean crossing contrasting environments . Results from this cruise are being valorized and some papers are already published. Moutin et al. (2017) described the hydrological and dynamical context of biogeochemical sampling. De Verneil et al. (2017) analyse a surface chlorophyll a bloom by combining in situ and remote sensing datasets. They characterize the role of the physical circulation, and in particular the role of surface mesoscale circulation responsible for the bloom's biogeochemical properties. This new paper wants to put the OUTPACE cruise into a more synoptic view by looking at surface satellite data in order to illustrate the role of large scale as well as meso/submesoscale dynamics on the biogeochemial/biological horizontal distributions of OUTPACE data. This is clearly a worthwhile exercise, and the authors have done a substantial work by compiling different satellite data set, and by using original diagnostics. The paper is well written, easy to read. The description of surface conditions by satellite data confirms most of the existing literature. But at first order, the motivation of this study is to do the link between this synoptic approach and the OUTPACE zonal section. It is not an easy task, and as a consequence most of the results are highly speculative. It is not really discuss what is the new contribution of this paper in comparison with the other existing papers dealing with the OUTPACE data. So, at this stage I am not convinced that the materials presented are strong enough to justify a publication. My general comment is to not recommend the paper for publication until an effort was done to better give evidence of the results that motivate this paper.**

The first objective of our study was to replace the in situ observations collected during the OUTPACE cruise into a synoptic view. In the framework of the OUTPACE special issue, the idea was also to provide, to everyone working on the same dataset, the most complete picture of the horizontal circulation as possible. We agree on the speculative

aspect raised by the reviewer, but we wanted to highlight the significance of each scale (from large to submesoscale) to fully understand the variations that biogeochemists or biologists could identify on their dataset collected during the cruise. Besides the descriptive characteristic of the paper, we feel it is important to give some ideas (speculative by definition) of what could be the effect of the different scales studied in this paper. The originality of this work includes: i) the descending approach in describing both large to meso/submesoscale circulations in the context of an oceanographic cruise; ii) the focus on horizontal submesoscale, as it was indeed already shown that it was difficult to find vertical submesoscale evidences (de Verneil et al., 2017); iii) the example of fine-scale physical influence on horizontal ecological distributions. We consider that these points are appropriate contributions for the community of the OUTPACE cruise and beyond, and we agree to let this appear with more clarity in our manuscript. In particular, we modified the section Introduction to better introduce our objectives at the beginning of the manuscript. We add in Section Introduction:

"In this study we replace the in situ observations collected during the OUTPACE cruise into a synoptic view of the WTSP circulation at different horizontal scale. We investigate through a descending approach the large, meso- and submesoscale patterns using in situ observations obtained during the OUTPACE cruise, coupled with satellite data. Remote sensing provides daily physical and biological information over the entire WTSP for a time period covering the cruise duration and beyond (from June,1 2014 to May,31 2015). The inter-comparison between physical lagrangian diagnostics and available biogeochemical/biological measurements explores the potential influence of each scale on biogeochemical variations. In particular, we propose to focus on the possible impact of horizontal small-scale ocean circulations on horizontal dispersal of tracers such as temperature, salinity or chlorophyll, as well as on biological dynamics. Two original case studies are also presented to illustrate the fine-scale physical influence on horizontal ecological distributions."

**I.4: "several studies indicate a strong mesoscale variability due to barotropic instabilities and the interactions of the major currents and jets with the numerous**

**islands of the region (Qiu et al., 2009; Hristova et al., 2014)." Barotropic insta-
bility doesn't appear as the only source of instability in the WTSP. If barotropic
instabilities are related to the NVJ/CSCC/NCJ system (at 16 âŮẹ S) in most places
baroclinic instabilities prevail as in the STCC-SEC system (Qiu and Chen, 2004;
Hristova et al., 2014).**

We agree with the reviewer and the sentence needs to be modified. We change our
text in p3 L4 to: "Superimposed on these large scale patterns, several studies indicate
a strong mesoscale variability due to barotropic, baroclinic instabilities and the interac-
tions of the major currents and jets with the numerous islands of the region (Qiu and
Chen, 2004; Qiu et al., 2009; Hristova et al., 2014)."

**General comment: A great number of satellite data are used. Four different
altimetry-derived velocity product are tested: The classic Ssalto/Duacs product
that provides maps of absolute geostrophic velocities at 1/4 âŮẹ of resolution, a
similar product at higher resolution (1/8 âŮẹ ) that may include an Ekman compo-
nent, and a cyclogeostrophy correction. The authors identify the last product as
the most accurate in situ surface currents with regards to the trajectories of the
SVP floats launched during OUTPACE. It is not surprising than adding the Ekman
component improve the comparison with surface drifters. The cyclogeostrophy
correction must improve the current associated with eddies by taking account
centrifugal acceleration. In my knowledge, it is a really new altimetric product
and it could be interesting to illustrate the improvement brought by this correc-
tion. On this part, I am reserved because we don't know what are the respective
contributions of high resolution, Ekman component, cyclogeostrophy. The re-
sult is based by looking at trajectories of numerical particles launched at only
the three particular locations where SVP floats were deployed. Besides the rel-
ative limited number of locations for the validation, it is difficult to validate such
conclusion based only on Fig. A1. If the LDC plot provides a good qualitative
agreement, it is not the case for LDA and LDB.**
As the choice of the best product relies on a qualitative comparison, we agree that it would be easier for the reader to actually analyze the same figures as we did it. In Fig. 1, we present the 8-days trajectories of in situ floats and numerical particles at each long-duration station and for each satellite-derived products considered (Geostrophy $1/4°$; geostrophy $1/8°$; geostrophy and Ekman $1/8°$; geostrophy, Ekman and cyclo-geostrophy $1/8°$). In the case of station LDA, none of the products shows a significant improvement but this can be due to the relative closeness of the station position to the New Caledonia coast. Indeed, we know that satellite measurements are not well resolved close to the coast, and especially near New Caledonia where the topography and bathymetry are very complex. In the cases of LDB and LDC, the increase in resolution does not modify the general pattern of the trajectories, but when adding the Ekman component, we can notice an improvement in the direction of the numerical particles. Even if the particle positions are offseted, their direction are consistent with those of in situ drifters. Cyclogeostrophy seems to accelerate the particles' displacements, which is not surprising. The final positions of the numerical particles are closest to the final position of in situ drifters in the case of LDC. In the context of the OUTPACE cruise, we consider that LDC is a good example of the contribution of each improvement parameter as in the case of LDA and LDB neither improvement or decay are clearly obvious on the trajectories. This analysis and figure appear in a new subsection *2.4 Comparison of satellite products with in situ drifters* as Anonymous Referee #1 also pointed out to the lack of explanations about the comparison with the different satellite datasets. A more quantitative comparison would need a larger dataset of surface floats and a statistic on a greater number of trajectories, but it is clear that such a work is out of the goal of the present study and of the topic of this journal.

Therefore we propose to add the following text into a new subsection 2.4:

"The choice of the satellite product that best represents the surface dynamics relies on a qualitative comparison between the trajectories of in situ floats launched during the OUTPACE cruise (see Section 2.1) and the trajectories of numerical particles computed with each of the satellite-derived velocity field described in Section 2.2. Figure

2 shows the 8-days trajectories of in situ floats and numerical particles at each long-duration station and for each satellite-derived products considered (Geostrophy 1/4°; geostrophy 1/8°; geostrophy and Ekman 1/8°; geostrophy, Ekman and cyclogeostrophy 1/8°). The comparison is restricted to 8 days for a better visualisation and to be consistent with duration of the LD stations. In the case of station LDA, none of the products displays a significant improvement of numerical trajectories. This lack of refinement between the different products may be due to the lack of accuracy of satellite products when getting closer to the coast. Indeed, satellite measurements are not well resolved close to the coast, and especially near New Caledonia where the topography and bathymetry are very complex. In the cases of LDB and LDC, the increase in resolution does not modify the general pattern of the trajectories. However when adding the Ekman component, we can notice an improvement in the direction of the numerical particle trajectories. Even if the particle positions are offseted, their direction are consistent with those of in situ drifters. Cyclogeostrophy seems to accelerate the particles' displacements. The final positions of the numerical particles are closest to the final position of in situ drifters in the case of LDC. This observation is not surprising considering that cyclogeostrophy represents the centrifugal acceleration. In the context of the OUTPACE cruise, we consider that LDB and LDC examples illustrate clear improvements of the new satellite product including geostrophy, the Ekman component and cyclogeostrophy. In the case of LDA and LDB neither clear improvement or decay are obvious on the trajectories. Moreover when considering the surface circulation, it also remains important to take the wind effect, through the Ekman component, into account as it will strongly influence the trajectories of surface waters at large, meso- and submesoscale. As most of the diagnostics used in this study are calculated through particle trajectory computations, the Ekman component is of major significance. As a consequence and to stay consistent all along this study, we decide to chose the product combining geostrophy, the Ekman component and cyclogeostrophy at 1/8° resolution to compute every diagnostics. In the following we will refer to this product as the total surface altimetry-derived velocity field."

Figure A1 have thus disappeared from Appendix.

**p.6 l.29: "It is clearly more interesting to use CLS data instead of commonly used AVISO geostrophic surface currents because they include the wind effect and cyclogeostrophy with higher resolution, as well as better represents in situ data." Okay, I agree that the last product should represent in a better way the surface circulation. The authors argue that it is the most interesting data set. I am not against this opinion but it depends on the objective. Is it really interesting to add an Ekman component when you are interested by mesoscale features? The Ekman velocity is a strong component of the surface circulation but it is limited to the thin Ekman layer. So what is the importance of transport in such layer with regard to OUTPACE objectives where processes on the vertical must be crucial and partly related to submesoscale features? In my opinion, it lacks a discussion on the interest of such product for your objective.**

In this study, we are interested in the surface circulation, from large to submesoscale, and it is very important to take into account the wind effect on the surface because, as we have seen earlier, it can modify the trajectories of surface waters or buoyant material in the surface layer. We also agree that vertical processes can be crucial but here we decided to focus this paper on the horizontal influence. Hence considering the Ekman component is of major significance as it will strongly influence the large but also the meso/submesoscale circulations. Even if it does not significantly modify the mesoscale structures (defined as eddies), it will affect the trajectories of the water masses/particles. In particular the Lyapunov exponents, that allow for the identification of physical barriers, are calculated through particle trajectory computations. The trajectories bias due to the wind effect addition will increase the more the time scale considered will be large. That is also why it is important to be consistent by using the same product to compute the different diagnostics at the different time/space time scales. One of the main focus of the OUTPACE cruise is to study the dynamics of the diazotrophs in the surface layer (a few dozen meters). The Ekman layer include

this typical layer on which, of course, vertical biological processes occur. Hence in this study, we consider the surface circulation in an homogeneous layer in which there are some vertical processes. We agree to add this discussion in the additional section (2.4) about satellite product comparison proposed in the previous comment: "When considering the surface circulation, it also remains important to take the wind effect, through the Ekman component, into account as it will strongly influence the trajectories of surface waters at large, meso- and submesoscale. As most of the diagnostics used in this study are calculated through particle trajectory computations, the Ekman component is of major significance."

**p.7 l.1: Figure 4 is cited before Fig. 3**

We are sorry about this mistake that would be corrected in a revised manuscript.

**Table 1: For me it is mysterious why the statistics of meanders vary so much between backward/forward experiments. Any explanation?**

The long-duration stations were positioned within mesoscale structures in real time during the course of the cruise (Moutin et al., this issue), which can explain a high rate of meanders. Indeed, depending on the dynamics of such structures a high number of particles can remain trapped inside or not. The stations were characterized at different stage of the structures' life. In particular, LDB and LDC were performed in structures that were coherent for a long time but that disappeared (by filamentation and by merging repesctively) quite rapidly after the stations occupation.

**General comment: The authors have used Lagrangian approaches based on the recent LAVD method, and the combination of OW and RP parameters used in d'Ovidio (2013). They are very interesting methodologies as they ensure that eddies correspond with trapped waters. It seems more robust that the more classic linear parameter used in Chelton et al. (2011). As it is written both methods are in good agreement so what is the advantage to mix both methods? There is always some subjectivity when defining eddies, also when looking at Fig. C2,**

**it seems that the RP contours are more in agreement with the velocity field. If one result of this paper is to test different methods to detect and track eddies, it could be interesting to test the methods classically used based on sea level extrema (Chelton, 2011; Chaigneau et al., 2009, Isern Fontanet, ).**

As highlighted by the reviewer, both methodologies ensure to detect trapping eddies which is the main criteria chosen by the authors in order to focus on structures that may have a biogeochemical/biological influence through mesoscale features' transport. Both methods are compared to check their consistency as, to our knowledge, it was never done before and the LAVD method is rather new. We agree that the eddy detection and tracking is a really interesting question. But we believe that comparing all different methodologies, as in Souza et al. (2011), is a large study that would require to test many different methods, on longer time scale than the cruise and, possibly, in different regions. That is not the objective of this work in the context of the OUTPACE cruise and the special issue, although we think it is a necessary work that should be performed in an other paper. The main objective of our study on mesoscale features is to give an overview of the number of features that may strongly influence the water masses transport and their general circulation in the context of the OUTPACE cruise. However to clarify why we use the three different methodologies (LAVD method, OW parameter and RP parameter), we add the following sentence in Section 3.2: "To ensure that this new detection method is consistent with previous approaches that identify mesoscale features (eulerian Okubo-Weiss parameter) or retention areas (RP), we compare the structures detected with both parameters."

**FSLE results are compared with surface gradient measured from the in situ OUTPACE data. Because the OUTPACE section is zonal, the corresponding surface gradient is mainly representative of cross track fronts. This aspect that limits the comparison of the two datasets is not mentioned.**

Indeed, this is a good point raised by the reviewer that is mentioned in a revised manuscript in Section 3.3: "The zonal characteristic of the OUTPACE section forces

the surface gradient identified with the TSG to be mainly representative of cross track fronts."

**p.8 l7: "using altimetric..including the wind effect" and the cyclogeostrophy correction?**

Yes, all the diagnostics discussed in section Results and Discussion are computed with the product that include an increase in resolution, geostrophy, the wind effect (Ekman) and cyclogeostrophy. We now give better attention to clarify this point in the manuscript. We believe that adding the new subsection 2.4, about the comparison of satellite products (as previously proposed) will/could help clarifying this aspect with the following sentence: "As a consequence and to stay consistent all along this study, we decide to chose the product combining geostrophy, the Ekman component and cyclogeostrophy at $1/8°$ resolution to compute every diagnostics."

**P.8 l.14: "or an artefact.. due to the short averaging.." It seems in contradiction with the sentence in section 2.3.1: ".. these simulations ensures that the use of a one year time period doesn't significantly modify statistical outputs". Also, with a RT of 15 days max for eddies it would be surprising that mesoscale dynamics have a strong influence on the mean circulation.**

We removed this sentence.

**p.8 l.15: "The meridional transport does not correspond to any surface current but is mainly due to the south-easterly trade winds. This meridional component appears due to the addition of the Ekman component in altimetry surface velocities (Fig.B1, Supplementary Material). The large scale transport of surface waters in the OUTPACE area is thus a combination of the transport by general well-known surface currents and wind- driven circulations." What are the dynamics which refer to these "surface currents"?? This sentence is a little bit ambiguous. If the "surface current" refers to geostrophic balance, it is normal that it doesn't take account for the well-known poleward Ekman transport. Now**

**if we look at the circulation inferred from Sverdrup theory, it is not so different of the streamfunction in Fig.2 despite its depth integrated estimation.**

Surface currents refer to general geostrophic currents, that can also be depth integrated, identified in the literature (Tomczak and Godfrey, 2013; Kessler and Cravatte, 2013; Ganachaud et al, 2014). Here we try to differentiate between the main currents described in the literature from the actual transport of waters. For example, it is well-known that the South Equatorial Current (SEC) flows westward but it is not obvious for every future reader that the transport induced by the wind is poleward, so that in total the waters would have a southwestward deviation when flowing into the SEC. We were trying to keep in mind that we address a paper to a large community of oceanographers and not only to physicists.

Moreover we have few example of transport calculation in the area. Tomzcak and Godfrey (2013) showed a streamfunction calculated with wind stress (Fig 4.4-4.7) that highlight a global westward transport with very little meridional transport (from the north to the south), apart when they clearly detected a southward flow corresponding to the western boundary current, the East Australian Current (EAC). But this transport occurs very close to the Australian coasts, whereas in our manuscript, we show that we have a southward flow occurring in the entire WTSP. We thus demonstrate that the surface transport can be slightly different from the well-known integrated transport. Kessler and Cravatte (2013) also show the Sverdrup transport stream function (Sv) calculated from Godfrey's Island Rule and the wind stress curl field in the Coral Sea (Figure 4(c)). We find the same southward transport at 10 S due to the South Equatorial Counter Current. However south of 10S, the transport is mainly westward with no southward component. We believe this is an important point for biogeochemists to know that the major part of the waters sampled during the cruise have a northern origin.

Nonetheless we understand the reviewer's concern about this part and the clarifications discussed here should appear in the subsection *3.1 Large scale wind-driven pathways*: "Very few studies got interested in the surface transport inferred by the wind in the WTSP. Indeed Tomczak and Godfrey (2013) calculated a streamfunction from windstress and show a global westward transport with very little meridional transport except in the EAC which is very close to the Australian coasts. Kessler and Cravatte (2013) also computed the Sverdrup transport streamfunction calculated from Godfrey's Island Rule and the wind stress curl field in the Coral Sea. We agree on the southward transport at 10S due to the SECC. However south of 10S, they only identify a westward transport."

**P.8 l18: "surface waters travel from northeast to southwest" Fig. C1 shows that most of the waters of importance for OUTPACE comes from the north and the northest as written in l.22.**

This point is right: the major part of the waters reaching the OUTPACE area comes from the north. We also want to highlight the eastern origin which is less important but cannot be neglected. The sentence p8ăL18 is changed as followsĂă: "Most of the surface waters travel southwest from the northern Ariane section, with a significant part that originates from northeast".

**p.8 L27; "west of 170 w" → East of??**

Indeed the ultra-oligotrophic waters are located east of 170W. We corrected this mistake.

**P.8 28: ".The wind-driven surface transport highlighted here could participate in the biogeochemical variations between western and eastern waters. Indeed the path through the Melanesian area may enrich these waters due to the contact with multiple islands whereas waters that directly recirculate within the gyre keep their ultra-oligotrophic characteristics. " The authors here want to do a link between the wind driven circulation at the surface and biogeochemical variations. It seems very speculative because their argument is the path through the Melanesian area that is also valid for what is named "surface current". Also, what is the role of these surface waters in the biochemical properties against deeper water and vertical processes??**

Here we want to highlight that the 170 W longitude limit between waters that recirculate within the gyre (eastward transport at 25S) and waters that flow directly south or southwestward match with the limit between oligotrophic (west of 170W) and ultra-oligotrophic (east of 170W) waters as shown in Moutin et al. (2017, this issue). We believe this may not be a simple coincidence but we agree that this assumption is very speculative. However de Verneil et al. (2017, this issue) showed that the meso/submesoscale horizontal circulation in this area of transition (170W) is a major component to explain the origin of a phytoplanktonic bloom. We thus suggest that this specific surface transport through the islands may create particular conditions in surface biogeochemical concentrations that can influence biological activity.

**p.9 l.4: "the OUTPACE cruise took place during an El Niño phase but they determined that climatological effects, upon the results of the cruise, were minimized." The trade winds are very sensitive to ENSO conditions in the WTSP. So the authors argue for little effects of the wind driven circulation upon the results of the cruise. So, it seems to be in contradiction with the fact to use the altimetric product including the wind effect. Also, the authors argue the interest to investigate the mesoscale circulation. I am not sure that the altimetric product they used is well suited for such purpose. At first order, meso and submescale are driven by internal dynamics.**

The OUTPACE cruise clearly took place during an El Niño phase as mentioned by Moutin et al. (2017, this issue). If the climatological effects can be minimized for biogeochemical sampling, we show that the circulation/transport is still strongly influenced by the trade winds. It is not surprising that the wind driven circulation plays a key role in this region. That is also why we decided to still take the wind effect in consideration. We added this clarification in p9 L4: " were minimized, for biogeochemical samplings. However we show that the circulation and transport is still strongly influenced by the trade winds." We agree with the reviewer that meso/submesoscale are driven by internal dynamics but the same confusion as raised earlier appears: when we talk about

meso and submesoscale circulation we mean in terms of trajectories at meso and sub-mesoscale not in terms of structures. It is clear that the addition of the wind modifies fine scale trajectories. It is thus important to take this effect into account when considering the backward and forward mesoscale trajectories or the detection of physical barriers. Fig. 2 and 3 are example of the difference between submesoscale features detected with FSLE calculated without and with the wind effect. We can notice that the features detected with geostrophy are still there when adding the wind effect but their shape are somewhat different at fine scale. So it is important to take these variations when considering fine scale distributions of the surface biogeochemical properties.

We believe that the additional subsection 2.4 brings a clarification on this aspect: "When considering the surface circulation, it also remains important to take the wind effect, through the Ekman component, into account as it will strongly influence the trajectories of surface waters at large, meso- and submesoscale. As most of the diagnostics used in this study are calculated through particle trajectory computations, the Ekman component is of major significance."

**p.9 l. 20: "If the major part propagates westward, the meridional band between 180W and 170W is identified as a region with mostly eastward propagation of mesoscale structures." Based in Fig.3, it is very hard for me to see propagation. This result is developed in the next sentences to argue for the importance of mesoscale dynamics to transport enriched-fluid into nutrient-poor gyre waters. But this eastern propagation is not really shown and as said by the authors there are no in situ data to illustrate this point. At this stage the discussion is highly speculative.**

As raised by the reviewer, this discussion lies on few observations, in particular in de Verneil et al., 2017, that remains to be proved by further in situ observations and analysis. We propose to take this part out to avoid speculations.

**P.10 l. 25 " enhanced by the mesoscale transport" Fig.4 shows an eastward transport at LDB but the link with mesoscale is not obvious. Are there particles**

**trapped into eddies that propagate eastward until LDB? And what their retention time and their distance travelled?**

When we talk about mesoscale we refer to the circulation at a scale of the order of 10-100 km which is different than the mesoscale coherent features. A specific experiment was computed around station LDB in de Verneil et al., 2017 to follow the individual trajectories of particles initialized inside the bloom. If the particle trajectories followed the physical boundaries detected with FSLE, no characteristics of eddy trapping has been identified at first sight.

**P.11 l. 7 "correlations" It is not really a correlation here**

Actually we calculated correlation coefficient between satellite products (SST and Chl) and in situ data from TSG. This comparison resulted in a reasonable correlation of 0.8 between in situ measurements and co-located satellite data as described in Section 2.2.

**P.11 l.25; "These latter results also exhibit that an FSLE existence does not necessarily create a gradient but probably needs pre-existing tracer gradients and a lifetime longer than few days." The orientation of the front against the direction of the density gradient could be also an explanation?**

This point is correct. The orientation of the front with respect to the direction of the density gradient is also a factor that controls the generation of a strong gradient. A cross-front gradient should be sharper than an along-front gradient. We propose to add this comment in Section 3.4: "Moreover the orientation of the front with respect to the direction of the density gradient is also a factor that controls the generation of a strong gradient."

**Fig.6. It should be better if the axis in Fig.6b is in âŮȩ (as in Fig. 6a) despite than in km. Section 3.4 is the section that best fits with the objective of the paper highlighted in the title. It shows two interesting case studies of interaction**

**between fronts and biogeochemical properties. In my opinion, it is the most interesting part of the paper but it is only 1 page. It is regrettable that such results are not discussed with regard to the results of De Verneil et al. (2017). Do strong density gradients correspond with the FSLEs discussed LDB and LDA?**

We appreciate the reviewer's interest about this part. As suggested (also by Anonymous Referee #1), the figures 6b and 7b are modified to plot the axis in degree (Fig. 4 and 5). De Verneil et al. (2017) described the fate of the bloom over the period of the station on the vertical and studied the possible origins of this bloom (vertical vs horizontal submesoscale processes). Based on the results of de Verneil et al. (2017), we can conclude that the LDB bloom was generated by horizontal submesoscale processes and bounded by some physical features acting like barriers. In this paper, we support the significance of horizontal submesoscale processes in driving the distribution of phytoplanktonic community. Moreover we add complementary information about the surface community structure of the bloom. Horizontal submesoscale circulation and features, not only help generating the bloom but also drive the species distribution in it. We believe this result is in agreement with what de Verneil et al. (2017) observedÂă: the horizontal submesoscale processes play an important role in the bloom's generation and evolution.

Multiple strong density gradient (as defined in Section 2.3.2) are identified on both High Frequency (HF) sampling transects (Fig. 6 and 7 see red crosses). Most of the strong gradient are localized on an FSLE structure for HFA whereas only few of them match with a FSLE structure in the case of HFB. This observation is in agreement with the discussion p13 l20Âă: the presence of a FSLE does not necessarily imply a density gradient and, vice versa, a density gradient does not necessarily need a FSLE structure to exist. However, it is encouraging that the FSLE highlighted by the comparison with phytoplankton abundances also match with a strong density gradient.

Following the reviewer's comment we add a discussion with de Verneil et al. (2017, this issue) in Section 3.4Âă: "de Verneil et al. (2017) identified the influence of the submesoscale processes in generating the bloom. They also showed that the bloom

was bounded by some physical features acting like barriers. The results previously presented in this paper support the significance of horizontal submesoscale processes in driving the bloom's dynamic but we also show that the distribution of phytoplanktonic community inside the bloom is conditioned by submesoscale features."

**References:**

de Verneil, A., Rousselet, L., Doglioli, A. M., Petrenko, A. A., Moutin, T. (2017). The fate of a southwest Pacific bloom: gauging the impact of submesoscale vs. mesoscale circulation on biological gradients in the subtropics. Biogeosciences, 14(14), 3471.

Ganachaud, A., Cravatte, S., Melet, A., Schiller, A., Holbrook, N. J., Sloyan, B. M., ... Ridgway, K. (2014). The Southwest Pacific Ocean circulation and climate experiment (SPICE). Journal of Geophysical Research: Oceans, 119(11), 7660-7686.

Kessler, W. S., Cravatte, S. (2013). Mean circulation of the Coral Sea. Journal of Geophysical Research: Oceans, 118(12), 6385-6410.

Moutin, T., Doglioli, A. M., De Verneil, A., Bonnet, S. (2017). Preface: The Oligotrophy to the UlTra-oligotrophy PACific Experiment (OUTPACE cruise, 18 February to 3 April 2015). Biogeosciences, 14(13), 3207.

Souza, J. M. A. C. D., De Boyer Montegut, C., Le Traon, P. Y. (2011). Comparison between three implementations of automatic identification algorithms for the quantification and characterization of mesoscale eddies in the South Atlantic Ocean. Ocean Science, 7(3), 317-334.

Tomczak, M., Godfrey, J. S. (2013). Regional oceanography: an introduction. Elsevier.

|  | LDA | LDB | LDC |
|---|---|---|---|
| Geostrophy 1/4° | | | |
| Geostrophy 1/8° | | | |
| Geostrophy + Ekman 1/8° | | | |
| Geostrophy + Ekman + Cyclogeostrophy 1/8° | | | |

**Fig. 1.** Trajectories of numerical particles computed with Ariane (purple) with each satellite products and in situ floats (colors) for 8 days after the starting date of each long-duration station (LDA, LDB LD

[Figure]

**Fig. 2.** FSLE features calculated with the high-resolution (only) geostrophy productÂă

[Figure]

**Fig. 3.** FSLE features calculated with the high-resolution geostrophy, Ekman and cyclogeostro-phy product.

**Fig. 4.** Modified figures 6b

[Figure]

Legend:
- Picoalgae x 100
- HNA
- LNA
- Bacteria
- Prochlorococcus

**Fig. 5.** Modified figures 7b

[Figure]

**Fig. 6.** Sea surface density calculated with TSG temperature and salinity (TEOS-10 standards) at the location of high frequency sampling during LDA (left). Red crosses show the position of stro

[Figure]

**Fig. 7.** Sea surface density calculated with TSG temperature and salinity (TEOS-10 standards) at the location of high frequency sampling during LDB (right). Red crosses show the position of stro